# FROM TOKENS TO NODES: SEMANTIC-GUIDED MOTION CONTROL FOR DYNAMIC 3D GAUSSIAN SPLATTING

**Jianing Chen**[1,2]\* **Zehao Li**[1,2]\* **Yujun Cai**[3]**, Hao Jiang**[1,2]†**, Shuqin Gao**[1]
**Honglong Zhao**[1]**, Tianlu Mao**[1,2]**, Yucheng Zhang**[1,2]†

[1]Institute of Computing Technology, Chinese Academy of Sciences, ICT
[2]University of Chinese Academy of Sciences, UCAS
[3]The University of Queensland
{chenjianing23s, jianghao}@ict.ac.cn

## ABSTRACT

Dynamic 3D reconstruction from monocular videos remains difficult due to the ambiguity inferring 3D motion from limited views and computational demands of modeling temporally varying scenes. While recent sparse control methods alleviate computation by reducing millions of Gaussians to thousands of control points, they suffer from a critical limitation: they allocate points purely by geometry, leading to static redundancy and dynamic insufficiency. We propose a motion-adaptive framework that aligns control density with motion complexity. Leveraging semantic and motion priors from vision foundation models, we establish patch-token-node correspondences and apply motion-adaptive compression to concentrate control points in dynamic regions while suppressing redundancy in static backgrounds. Our approach achieves flexible representational density adaptation through iterative voxelization and motion tendency scoring, directly addressing the fundamental mismatch between control point allocation and motion complexity. To capture temporal evolution, we introduce spline-based trajectory parameterization initialized by 2D tracklets, replacing MLP-based deformation fields to achieve smoother motion representation and more stable optimization. Extensive experiments demonstrate significant improvements in reconstruction quality and efficiency over existing state-of-the-art methods.

## 1 INTRODUCTION

Dynamic 3D reconstruction from monocular videos is critical for virtual reality, autonomous systems, and content creation. The task requires capturing complex object motions and deformations from limited viewpoints while maintaining real-time rendering performance. This remains challenging due to the fundamental ambiguity of inferring 3D motion from 2D observations and the computational demands of modeling temporally varying scenes.

Recent advances in 3D Gaussian Splatting (Kerbl et al. (2023)) have enabled efficient static scene reconstruction through explicit primitive representations and fast rasterization. Extensions to dynamic scenes follow two approaches: dense methods that parameterize each Gaussian's temporal evolution, achieving high quality but poor scalability, and sparse control methods that use a small set of control points to govern scene deformation. Sparse approaches like SC-GS (Huang et al. (2023), SP-GS Diwen Wan (2024)) and 4D-Scaffold (Cho et al. (2025)) offer significant computational savings by reducing the optimization space from hundreds of thousands of Gaussians to thousands of control points.

However, existing sparse methods suffer from a fundamental limitation: they allocate control points based purely on geometric considerations. Methods typically use Farthest Point Sampling (FPS)

---

\*Equal contribution
†Corresponding authors

(Huang et al. (2023); Diwen Wan (2024); Chen et al. (2025)) or voxel centers (Cho et al. (2025); Kong et al. (2025)) to ensure uniform spatial coverage, but this geometric uniformity does not align with motion complexity. Real scenes exhibit highly non-uniform motion where static backgrounds dominate spatial extent while dynamic objects occupy smaller regions but require detailed motion modeling. This mismatch leads to **static redundancy yet dynamic insufficiency**, where control points are wasted on static regions while dynamic areas remain under-represented.

We address this through motion-adaptive control point allocation guided by vision foundation models. Our approach is built on the insight that semantic understanding can predict motion patterns: certain object categories exhibit predictable motion behaviors that can be learned from large-scale video datasets. We leverage pre-trained vision foundation models to extract semantic tokens from image patches and establish patch-token-node correspondence, enabling direct transfer of 2D semantic priors to 3D control point placement.

Our method operates in three stages. First, we generate candidate nodes by back-projecting image patches into 3D space using estimated depth and camera poses, with each node retaining its semantic token as a descriptor. Second, we apply motion-adaptive compression that iteratively merges nodes based on semantic similarity and motion tendency scores derived from vision foundation models. This compression concentrates control points in dynamic regions while reducing redundancy in static areas, directly addressing the static-dynamic resource allocation mismatch. Third, we parameterize node trajectories using cubic splines rather than MLPs, initialized from 2D tracklets to provide stable motion guidance during optimization. This spline formulation offers several advantages. It ensures temporal smoothness, reduces optimization complexity by decoupling trajectory learning from other parameters, and provides a compact representation that scales better than dense deformation fields.

In summary, our main contributions are:

- We propose a motion-adaptive node initialization method using semantic and motion priors from vision foundation models to align control density with motion complexity.

- We introduce a spline-based parameterization of node trajectories, which provides a compact, smooth, and differentiable motion basis for the entire dynamic scene.

- We present a complete optimization framework demonstrating superior reconstruction quality and efficiency over existing methods.

## 2 RELATED WORK

**Dynamic NeRF.** Neural Radiance Fields (NeRF) (Mildenhall et al. (2020)) pioneered static view synthesis via implicit volumetric MLPs. Subsequent works (Guo et al. (2023); Gafni et al. (2021); Park et al. (2021a;b); Pumarola et al. (2021); Fang et al. (2022); Wang et al. (2023)) extended NeRF to dynamic scenes with temporal structures such as deformation fields and canonical mappings, but remain inefficient due to dense ray sampling and costly volume rendering. To improve efficiency, recent methods introduce grid-based representations (Liu et al. (2022)) and multi-view supervision (Lin et al. (2022; 2023)), while explicit representations such as multi-plane (Chen et al. (2022); Fridovich-Keil et al. (2023b); Shao et al. (2023)) and grid-plane hybrids (Song et al. (2023)) further accelerate training. Nonetheless, their rendering speed is still insufficient for real-time applications.

**Dynamic Gaussian Splatting.** 3D Gaussian Splatting (3DGS) (Kerbl et al. (2023)) enables real-time rendering with explicit point-based representations and shows potential for broader 3D tasks (Li et al. (2024); Qu et al. (2024); Cai et al. (2019; 2020)). Recent works have extended 3DGS to dynamic scenes by learning time-varying Gaussian transformations. Several approaches (Yang et al. (2024b); Li et al. (2025)) adopt per-Gaussian deformation fields, but such designs often incur redundant computation and slow training. Later methods adopt compact structural representations, such as plane encodings or hash-based schemes (Wu et al. (2024); Xu et al. (2024)), to improve deformation efficiency. Alternatively, sparse control points have been introduced (Huang et al. (2023); Diwen Wan (2024); Kong et al. (2025); Lei et al. (2025); Chen et al. (2025); Liang et al. (2025)) as a lightweight mechanism to govern Gaussian motion via interpolation, supporting both high-quality rendering and motion editing. Existing approaches differ in how control points are initialized: SC-GS, SP-GS, and HAIF-GS (Huang et al. (2023); Diwen Wan (2024); Chen et al. (2025)) adopt FPS

sampling to ensure uniform spatial coverage, while 4D-Scaffold and EDGS (Cho et al. (2025); Kong et al. (2025)) use voxelization, which proves suboptimal in real-world scenes dominated by static backgrounds. More recent methods, such as MoSca and HiMoR (Lei et al. (2025); Liang et al. (2025)), leverage 2D tracklets from vision foundation models, but they remain sensitive to tracking errors and struggle with large topological variations. Despite these advances, sparse control methods still fail to adapt control density to motion complexity, often resulting in static redundancy and dynamic insufficiency. To address this, we propose a motion-adaptive 3DGS framework that re-allocates control points according to motion cues and further stabilizes trajectory learning through spline parameterization.

## 3 PRELIMINARY: 3D GAUSSIAN SPLATTING

3D Gaussian Splatting (3DGS) (Kerbl et al. (2023)) models a static scene as anisotropic 3D Gaussians, each parameterized by center $\mu \in \mathbb{R}^3$, covariance $\mathbf{\Sigma} \in \mathbb{R}^{3 \times 3}$, opacity $\alpha \in (0, 1)$, and spherical harmonics (SH) coefficients $\mathbf{c} \in \mathbb{R}^{3(l+1)^2}$ for view-dependent color, denoted as $G(\mu, \mathbf{\Sigma}, \alpha, \mathbf{c})$.

Each Gaussian is projected to the image plane through the camera projection, forming a 2D Gaussian that contributes to pixel colors. The 2D Gaussians are sorted by depth and rendered via an $\alpha$-blending scheme. The color at pixel $p$ is obtained by compositing the contributions of $N$ ordered Gaussians overlapping the pixel:

$$C(p) = \sum_{i \in N} \mathbf{c}_i \, \alpha_i \prod_{j=1}^{i-1} (1 - \alpha_j), \tag{1}$$

where $\mathbf{c}_i$ is the color of the $i$-th Gaussian and $\alpha_i$ is its image-space density determined by the projected covariance. The parameters are optimized with a photometric reconstruction loss, and adaptive density control dynamically prunes or spawns Gaussians to improve efficiency and fidelity.

Extending 3DGS to dynamic scenes is commonly formalized by endowing the representation with explicit temporal parameterization instead of a purely canonical configuration. Following prior work (Liang et al. (2025); Wang et al. (2024)), we introduce a temporal transformation that maps each Gaussian from the canonical space to its state at frame $t$, written as $\mathbf{T}_t = [\mathbf{R}_t \mid \mathbf{t}_t] \in \mathrm{SE}(3)$. Applying $\mathbf{T}_t$ to a canonical Gaussian $G(\mu_0, \mathbf{\Sigma}_0, \alpha, \mathbf{c})$ yields its time-varying form $G_t = G(\mathbf{T}_t \mu_0, \mathbf{R}_t \mathbf{\Sigma}_0, \alpha, \mathbf{c})$, which provides a compact parameterization of dynamic scenes.

## 4 METHOD

### 4.1 OVERVIEW

Given a monocular image sequence $\{I_t\}$, our goal is to reconstruct a dynamic 3DGS representation that enables temporally consistent, photorealistic novel-view renderings. The central challenge lies in the spatially non-uniform motion complexity and the need for smooth, stable trajectories under sparse supervision. To address this, we adopt a sparse node-based deformation representation that controls canonical Gaussians (Sec. 4.2) through motion-adaptive allocation. we first initialize nodes from image patches and leverage semantic and motion cues from vision foundation models to compress redundant nodes in static regions while preserving those in dynamic regions (Sec. 4.3). We then parameterize node trajectories with a spline to provide a compact, smooth, and differentiable motion basis, initialized from 2D tracklets for stable early-stage optimization (Sec. 4.4). Finally, we propagate node transforms to Gaussians through dual quaternion blending and jointly optimize geometry, appearance, and motion with multi-view photometric and motion-consistency constraints (Sec. 4.5). Figure 1 summarizes our pipeline, which integrates motion-adaptive compression with iterative voxelization to flexibly adapt representational density according to motion complexity.

### 4.2 NODE-BASED DEFORMATION REPRESENTATION

Modeling deformations in dynamic Gaussian scenes requires balancing expressiveness with tractability. Direct per-primitive formulations are prohibitively high-dimensional, while real-world motion often exhibits low-rank structure dominated by rigid and smooth patterns. This motivates a compact node-based representation, where each node carries an $\mathrm{SE}(3)$ trajectory and an RBF kernel defining its spatial influence. Gaussians inherit motion from their $K$ nearest nodes through weighted aggregation, forming an efficient basis for our subsequent initialization and trajectory modeling.

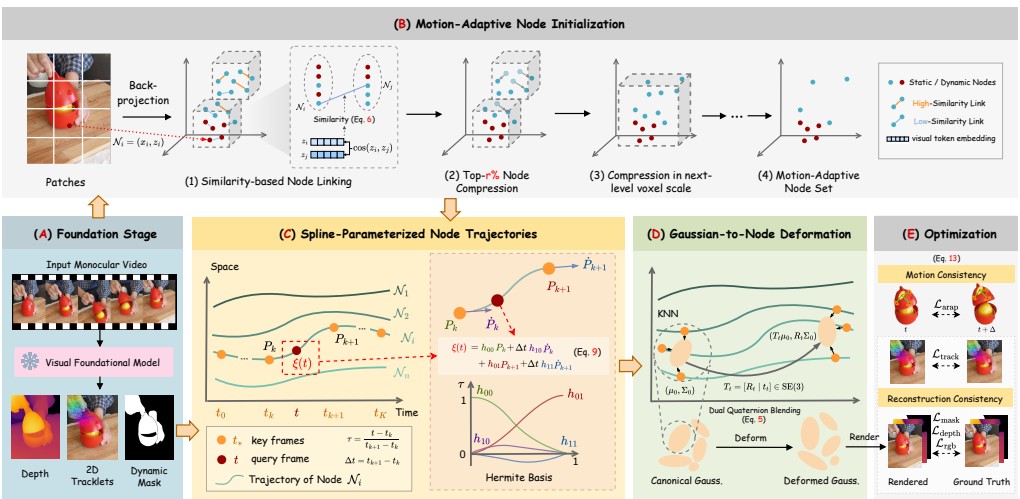

Figure 1. **The overview of our method.** (A) Given a monocular video, we extract semantic and motion priors from pre-trained vision foundation models. (B) These priors guide motion-adaptive node initialization, yielding compact distributions aligned with dynamic regions. (C) The initialized nodes are assigned spline-parameterized trajectories to provide a motion basis. (D) Node motions are propagated to Gaussians through deformation, transforming the canonical representation. (E) The deformed model is rendered and optimized for consistent reconstruction.

**Node Representation.** We introduce a sparse set of nodes $\mathcal{N} = \{\mathcal{N}_i\}_{i=1}^{N_n}$ to capture the dominant smooth motion patterns of the scene, where the number of nodes $N_n$ is significantly smaller than the number of Gaussian primitives $N_g$. Each node is formally defined as

$$\mathcal{N}_i = \{\mathbf{T}_i(t), \rho_i\}, \tag{2}$$

where $\mathbf{T}_i(t) \in \mathrm{SE}(3)$ denotes the trajectory of $\mathcal{N}_i$ across time, and $\rho_i \in \mathbb{R}^+$ specifies the radius of its radial basis function (RBF), which determines the spatial extent of its influence. Thus, $\mathbf{T}_i(t)$ governs rigid motion over time, while $\rho_i$ determines the spatial scope of influence. This node formulation further supports motion-adaptive initialization, allowing dynamic regions to be modeled with higher fidelity (Sec. 4.3). To ensure smooth and compact temporal modeling, each trajectory is parameterized by splines (Sec. 4.4).

**Gaussian-to-Node Binding and Deformation.** We derive the rigid transformation of each Gaussian primitive $\mathcal{G}_j$ at any query time $t$ by leveraging the trajectories of its neighboring nodes. Given the node set $\mathcal{N} = \{\mathcal{N}_i\}_{i=1}^{N_n}$, each Gaussian $\mathcal{G}_j$ is associated with a neighborhood of $K$ nodes, denoted $\mathcal{V}(G_j) \subset \mathcal{N}$. The binding weight of node $\mathcal{N}_i$ to Gaussian $\mathcal{G}_j$ is defined as

$$w_{ij} = \frac{\exp\left(-\frac{\|\mathbf{x}_j - \mathbf{c}_i\|^2}{2\rho_i^2}\right)}{\sum_{k \in \mathcal{V}(G_j)} \exp\left(-\frac{\|\mathbf{x}_j - \mathbf{c}_k\|^2}{2\rho_k^2}\right)}, \tag{3}$$

where $\mathbf{x}_j$ is the canonical center of Gaussian $\mathcal{G}_j$, $\mathbf{c}_i$ is the canonical center of node $\mathcal{N}_i$. These normalized weights act as interpolation coefficients in the blending stage.

To propagate node motion to Gaussians, we construct a dense deformation field that interpolates per-Gaussian rigid motions from sparse node trajectories. Following prior work (Lei et al. (2025)), we instantiate this field with Dual Quaternion Blending (DQB) (Kavan et al. (2007)), which provides better interpolation quality. Concretely, for a node $\mathcal{N}_i$, its $\mathrm{SE}(3)$ transform at time $t$ is written as $\mathbf{T}_i(t) = [\mathbf{R}_i(t) \,|\, \mathbf{t}_i(t)]$. Its dual quaternion representation $\mathbf{Q}_i(t) \in \mathbb{DQ}$ is constructed as

$$\mathbf{Q}_i(t) = q_{r,i}(t) + \epsilon \, q_{d,i}(t), \quad q_{d,i}(t) = \tfrac{1}{2} p_i(t) \, q_{r,i}(t), \tag{4}$$

where $q_{r,i}(t)$ is the unit quaternion corresponding to $\mathbf{R}_i(t)$, $p_i(t)$ is the pure quaternion of the translation vector $\mathbf{t}_i(t)$, and $\epsilon$ is the dual unit with $\epsilon^2 = 0$.

The blended transformation for Gaussian $\mathcal{G}_j$ is obtained by normalizing the weighted sum of neighboring nodes' dual quaternions and mapping the result back to SE(3):

$$\hat{\mathbf{Q}}_j(t) = \frac{\sum_{i \in \mathcal{V}(G_j)} w_{ij}\, \mathbf{Q}_i(t)}{\left\| \sum_{i \in \mathcal{V}(G_j)} w_{ij}\, \mathbf{Q}_i(t) \right\|}, \quad \mathbf{T}_j(t) = \mathrm{DQ2SE3}\!\left(\hat{\mathbf{Q}}_j(t)\right). \tag{5}$$

Here normalization guarantees that $\hat{\mathbf{Q}}_j(t)$ remains a unit dual quaternion, while $\mathrm{DQ2SE3}(\cdot)$ denotes the standard conversion from a unit dual quaternion to a rigid transform. This formulation enables Gaussian motion to be obtained through weighted blending of neighboring node trajectories, ensuring physical consistency and temporal smoothness.

## 4.3 MOTION-ADAPTIVE NODE INITIALIZATION

Building upon the node representation in Sec. 4.2, we now address how to initialize nodes in a way that adapts to motion complexity. Uniform sampling tends to oversample static backgrounds while failing to capture sufficient detail in dynamic regions, resulting in biased motion modeling. To overcome this imbalance, we introduce a semantic-guided, motion-adaptive initialization that allocates more nodes to dynamic areas while reducing redundancy elsewhere. Given calibrated keyframes with depth and semantics, this procedure generates a compact node set in canonical space that serves as the starting point for subsequent deformation modeling.

**Patch-to-Node Generation.** To better integrate semantic cues with geometry, we generate candidate nodes directly from image patches rather than uniformly sampling point clouds or voxelizing 3D space. Specifically, we select a set of keyframes $\{I_t\}_{t=1}^{T}$ and divide each image into fixed-size patches $\{p\}$. A frozen vision foundation model provides a token embedding $z_{t,p}$ for each patch $p$ at frame $t$, along with estimated depth maps. Each patch center $\mathbf{u}_{t,p}$ is back-projected into 3D space to obtain its coordinate $\mathbf{x}_{t,p}$. The resulting collection $\{(\mathbf{x}_{t,p}, z_{t,p})\}$ forms the initial candidate node set, where each node is anchored at the patch center and retains the semantic token as its descriptor. This preserves a patch–token–node correspondence that can be exploited during subsequent compression.

**Dynamic Motion-Adaptive Node Compression.** The candidate node set is still excessively large for direct modeling, necessitating a principled compression strategy. A naive voxelization with fixed resolution is insufficiently adaptive across regions and often mixes features of distinct objects. We therefore propose an iterative motion-adaptive compression that iteratively merges nodes while preserving fidelity in dynamic areas. Starting from a small initial voxel size $v_{\mathrm{init}}$, the voxel resolution is progressively enlarged during compression. In each iteration, bipartite soft matching (Huang et al. (2025)) is applied within every voxel. For each node in $A$, we connect it to the most similar node in $B$, and the top $r\%$ pairs with the highest similarity are merged by retaining one representative node. After completing all voxels in the current iteration, the voxel size is enlarged by a fixed step $\Delta v$, and the process is repeated until the node count falls below a target threshold.

To ensure that merging respects both appearance and geometry, we define a joint similarity between nodes $\mathcal{N}_i \in A$ and $\mathcal{N}_j \in B$ as

$$\mathrm{sim}(\mathcal{N}_i, \mathcal{N}_j) = \cos(z_i, z_j) - \eta \cdot \tilde{M}_{\mathrm{fg}}(\mathcal{N}_i, \mathcal{N}_j), \tag{6}$$

where $\cos(z_i, z_j)$ measures the token-based appearance similarity, and $\tilde{M}_{\mathrm{fg}}(\mathcal{N}_i, \mathcal{N}_j) \in [0, 1]$ denotes a foreground prior predicted by a frozen VFM. Tokens from VFMs encode both semantic context and local appearance. Static regions yield consistent tokens across views, whereas motion causes variations that lower their similarity. Thus, cosine similarity serves as an effective cue to distinguish dynamic from static areas. The mask prior provides coarse localization of dynamic areas, discouraging premature merging in regions with high dynamic likelihood.

However, simply applying a uniform compression ratio across all voxels fails to leverage this motion-aware similarity information effectively. Such uniform treatment leads to an unfavorable trade-off: a high ratio prematurely merges dynamic nodes during early fine-voxel stages, while a low ratio fails to sufficiently reduce redundancy in static regions. To address this limitation, we propose an adaptive compression strategy that adjusts the compression ratio according to the motion tendency of each voxel cluster. Concretely, we define a dynamic tendency score $p_{\mathrm{dyn}}(C)$ for a cluster $C$ by combining the mean foreground prior with the pairwise similarity within the cluster:

$$p_{\text{dyn}}(C) = \sigma \left( \alpha \cdot \frac{1}{|\mathcal{U}_C|} \sum_{\mathcal{N}_k \in \mathcal{U}_C} m(\mathcal{N}_k) - \beta \cdot \frac{1}{|\mathcal{M}_C|} \sum_{(\mathcal{N}_i, \mathcal{N}_j) \in \mathcal{M}_C} \text{sim}(\mathcal{N}_i, \mathcal{N}_j) \right), \qquad (7)$$

where $\mathcal{U}_C$ denotes the set of nodes in cluster $C$, and $\mathcal{M}_C$ the set of their matched pairs. This score is then used to modulate the compression ratio of each cluster:

$$r\%(C) = r_{\min} + (1 - p_{\text{dyn}}(C)) \cdot (r_{\max} - r_{\min}), \qquad (8)$$

so that static voxels with low $p_{\text{dyn}}$ are merged aggressively with a high $r\%$, while dynamic voxels with high $p_{\text{dyn}}$ are preserved with a low $r\%$.

In this way, compression reduces redundancy in static regions while maintaining sufficient node density in dynamic areas, striking a balance between efficiency and temporal modeling fidelity.

### 4.4 SPLINE-PARAMETERIZED NODE TRAJECTORIES

Given the motion-adaptive node set in the canonical space, the next challenge is to represent their temporal evolution. Directly optimizing node positions at every frame is unstable and computationally expensive, as it lacks temporal regularization and entangles motion learning with Gaussian attribute updates. To achieve sparse yet stable control, we parameterize each node trajectory with a small set of keyframes connected by cubic splines. This spline-based formulation enforces smooth and differentiable trajectories, alleviates early-stage optimization difficulty, and provides reliable motion guidance for the associated Gaussian primitives.

**Spline-Based Formulation.** To obtain the motion of each Node at arbitrary time steps, we represent its trajectory with a cubic Hermite spline (Park et al. (2025); Ahlberg et al. (2016); Goodfellow et al. (2016)). Concretely, we select a set of keyframes $\{t_k\}_{k=1}^K$ along the timeline and assign learnable positions $\{P_k\}_{k=1}^K$ to the Node at these frames. The trajectory $\xi(t)$ between two neighboring keyframes $(t_k, t_{k+1})$ is then interpolated as

$$\xi(t) = h_{00}(\tau)\, P_k + h_{10}(\tau)\, (t_{k+1} - t_k)\, \dot{P}_k + h_{01}(\tau)\, P_{k+1} + h_{11}(\tau)\, (t_{k+1} - t_k)\, \dot{P}_{k+1}, \quad (9)$$

where $\tau = \frac{t - t_k}{t_{k+1} - t_k}$, and the Hermite basis functions are

$$\begin{aligned}
h_{00}(\tau) &= 2\tau^3 - 3\tau^2 + 1, \quad h_{10}(\tau) = \tau^3 - 2\tau^2 + \tau, \\
h_{01}(\tau) &= -2\tau^3 + 3\tau^2, \quad h_{11}(\tau) = \tau^3 - \tau^2.
\end{aligned} \qquad (10)$$

This spline-based construction ensures temporal continuity by keeping both positions and first-order derivatives consistent across time. More importantly, it provides a compact and differentiable representation that avoids the instability and heavy joint optimization associated with MLP-based deformation fields, thereby offering stable guidance for the Gaussian primitives bound to these nodes.

**Trajectory Initialization.** To provide stable guidance at the early stage, we initialize the spline-parameterized node trajectories from geometry-consistency, instead of using random parameters. Concretely, we extract long-term 2D tracklets (Doersch et al. (2023)) from a sequence of frames, and unproject them into world coordinates using estimated depth (Piccinelli et al. (2024)) and camera poses. Formally, given a pixel coordinate $u_t$ on the 2D track at time $t$ with depth $D_t(u_t)$, its world-space position is computed as

$$x_t = \mathbf{R}_t^\top \pi_\mathbf{K}^{-1}\big(u_t, D_t(u_t)\big) - \mathbf{R}_t^\top \mathbf{T}_t, \qquad (11)$$

where $\pi_\mathbf{K}^{-1}(\cdot)$ denotes the back-projection from image to camera space with intrinsic $\mathbf{K}$, and $(\mathbf{R}_t, \mathbf{T}_t)$ are the estimated extrinsics. We then initialize the **translational** spline by fitting a Hermite trajectory $\xi(t)$, over keyframes $\{t_k\}_{k=1}^K$, to the 3D tracklets $\{x_t\}$ via least-squares optimization:

$$\min_{\{P_k\}_{k=1}^K} \sum_{t=0}^{N_f - 1} \big\| x_t - \xi(t) \big\|_2^2, \qquad (12)$$

where $\{P_k\} \subset \mathbb{R}^3$ denote the learnable node positions at the keyframes, and $\xi(t)$ between $(t_k, t_{k+1})$ follows the cubic Hermite basis described previously. For the **rotational** component, we initialize $\mathbf{R}^{\text{node}}(t) = \mathbf{I}_3$ for all $t$, and defer its refinement to the joint optimization stage.

Table 1. **Quantitative comparison** on Hyper-NeRF(vrig) dataset per-scene. We highlight the best , second best and the third best results in each scene.

| Method | Broom | | | 3D-Printer | | | Chicken | | | Banana | | | Mean | | |
|---|---|---|---|---|---|---|---|---|---|---|---|---|---|---|---|
| | PSNR↑ | SSIM↑ | LPIPS↓ | PSNR↑ | SSIM↑ | LPIPS↓ | PSNR↑ | SSIM↑ | LPIPS↓ | PSNR↑ | SSIM↑ | LPIPS↓ | PSNR↑ | SSIM↑ | LPIPS↓ |
| HyperNeRF Park et al. (2021b) | 19.51 | 0.210 | - | 20.04 | 0.635 | - | 27.46 | 0.828 | - | 22.15 | 0.719 | - | 22.29 | 0.598 | - |
| TiNeuVox Fang et al. (2022) | 21.28 | 0.307 | - | 22.80 | 0.725 | - | 28.22 | 0.785 | - | 24.50 | 0.646 | - | 24.20 | 0.616 | - |
| D-3DGS Yang et al. (2024b) | 19.99 | 0.269 | 0.700 | 20.71 | 0.656 | 0.277 | 22.77 | 0.640 | 0.363 | 25.95 | 0.853 | 0.155 | 22.36 | 0.605 | 0.374 |
| 4DGS Wu et al. (2024) | 22.01 | 0.366 | 0.557 | 21.98 | 0.705 | 0.327 | 28.49 | 0.806 | 0.297 | 27.73 | 0.847 | 0.204 | 25.05 | 0.681 | 0.346 |
| MotionGS Zhu et al. (2024) | 22.30 | 0.380 | - | 21.80 | 0.710 | - | 26.80 | 0.790 | - | 28.20 | 0.690 | - | 24.78 | 0.643 | - |
| MoSca Lei et al. (2025) | 22.14 | 0.414 | 0.415 | 22.26 | 0.691 | 0.245 | 28.19 | 0.817 | 0.199 | 28.43 | 0.866 | 0.170 | 25.25 | 0.697 | 0.257 |
| ED3DGS Bae et al. (2024) | 21.84 | 0.371 | 0.531 | 22.34 | 0.715 | 0.294 | 28.75 | 0.836 | 0.185 | 28.80 | 0.867 | 0.178 | 25.43 | 0.697 | 0.297 |
| MoDec-GS Kwak et al. (2025) | 21.04 | 0.303 | 0.666 | 22.00 | 0.706 | 0.265 | 28.77 | 0.834 | 0.197 | 28.25 | 0.873 | 0.173 | 25.02 | 0.679 | 0.325 |
| Grid4D Xu et al. (2024) | 21.78 | 0.414 | 0.423 | 22.36 | 0.723 | 0.245 | 29.27 | 0.848 | 0.199 | 28.44 | 0.875 | 0.176 | 25.46 | 0.715 | 0.261 |
| SC-GS Huang et al. (2023) | 18.66 | 0.269 | 0.505 | 18.79 | 0.613 | 0.269 | 21.85 | 0.616 | 0.257 | 25.49 | 0.806 | 0.215 | 21.20 | 0.576 | 0.312 |
| SC-GS+MANI | 19.93 | 0.284 | 0.491 | 20.61 | 0.653 | 0.255 | 23.20 | 0.684 | 0.230 | 26.88 | 0.823 | 0.207 | 22.66 | 0.611 | 0.296 |
| **Ours** | 22.37 | 0.421 | 0.405 | 22.53 | 0.729 | 0.232 | 29.66 | 0.863 | 0.161 | 28.55 | 0.879 | 0.168 | 25.78 | 0.723 | 0.242 |

Table 2. **Quantitative comparison** on N3DV dataset per-scene. We highlight the best , second best and the third best results in each scene.

| Method | Coffee Martini | | Cook Spinach | | Cut Beef | | Flame Salmon | | Flame Steak | | Sear Steak | | Mean | |
|---|---|---|---|---|---|---|---|---|---|---|---|---|---|---|
| | PSNR↑ | SSIM↑ | PSNR↑ | SSIM↑ | PSNR↑ | SSIM↑ | PSNR↑ | SSIM↑ | PSNR↑ | SSIM↑ | PSNR↑ | SSIM↑ | PSNR↑ | SSIM↑ |
| HexPlane Cao & Johnson (2023) | 13.26 | 0.405 | 16.95 | 0.729 | 16.76 | 0.538 | 11.16 | 0.342 | 16.97 | 0.753 | 16.89 | 0.589 | 15.33 | 0.559 |
| D-3DGS Yang et al. (2024b) | 19.23 | 0.701 | 17.20 | 0.720 | 22.20 | 0.780 | 18.48 | 0.704 | 16.62 | 0.752 | 23.56 | 0.810 | 19.55 | 0.745 |
| 4DGS Wu et al. (2024) | 20.95 | 0.761 | 22.64 | 0.779 | 23.18 | 0.793 | 20.64 | 0.758 | 21.83 | 0.787 | 23.38 | 0.829 | 22.10 | 0.785 |
| SC-GS Huang et al. (2023) | 19.02 | 0.712 | 16.70 | 0.737 | 20.69 | 0.741 | 17.65 | 0.683 | 17.31 | 0.753 | 21.23 | 0.787 | 18.77 | 0.736 |
| MoDGS Qingming et al. (2025) | 21.37 | 0.796 | 22.40 | 0.782 | 23.89 | 0.822 | 21.33 | 0.804 | 23.23 | 0.808 | 23.53 | 0.812 | 22.63 | 0.804 |
| Grid4D Xu et al. (2024) | 21.32 | 0.791 | 22.58 | 0.788 | 23.51 | 0.827 | 21.04 | 0.800 | 23.45 | 0.815 | 23.14 | 0.806 | 22.51 | 0.805 |
| **Ours** | 22.53 | 0.824 | 22.97 | 0.795 | 24.36 | 0.836 | 21.97 | 0.823 | 23.89 | 0.821 | 24.13 | 0.827 | 23.31 | 0.821 |

This geometry-driven initialization strategy grounds the spline trajectories in observed motion patterns, producing stable translational paths while preserving rotational flexibility, which facilitates more robust convergence during optimization.

### 4.5 OPTIMIZATION

To stabilize optimization under the monocular setting, we design a composite loss that integrates photometric, geometric, and motion-related constraints:

$$\mathcal{L}_{\text{total}} = \lambda_{\text{rgb}}\mathcal{L}_{\text{rgb}} + \lambda_{\text{mask}}\mathcal{L}_{\text{mask}} + \lambda_{\text{depth}}\mathcal{L}_{\text{depth}} + \lambda_{\text{track}}\mathcal{L}_{\text{track}} + \lambda_{\text{arap}}\mathcal{L}_{\text{arap}}. \tag{13}$$

The photometric loss $\mathcal{L}_{\text{rgb}}$ follows the standard practice in 3DGS (Kerbl et al. (2023)), encouraging rendered views to be consistent with the input images. The mask loss $\mathcal{L}_{\text{mask}}$ employs foreground masks predicted by an off-the-shelf segmentation model (Yang et al. (2023)) as supervision signals. The depth loss $\mathcal{L}_{\text{depth}}$ leverages relative depth maps estimated from a monocular depth prediction model (Hu et al. (2025)), aligned with sparse geometric priors to improve structural accuracy. For motion supervision, the tracking loss $\mathcal{L}_{\text{track}}$ enforces temporal consistency by constraining the projected motion of rendered points against trajectories obtained from a pre-trained 2D tracking model (Doersch et al. (2023)). Finally, the ARAP loss $\mathcal{L}_{\text{arap}}$ (Huang et al. (2024); Lei et al. (2025)) regularizes control point motion by penalizing non-rigid distortions in local neighborhoods, thereby ensuring locally rigid deformations and preventing unrealistic stretching. Detailed formulations of the above loss terms are provided in Appendix. A.2.

## 5 EXPERIMENTS

### 5.1 EXPERIMENTAL SETUP

**Datasets and Metrics.** We evaluate our method on two real-world datasets: Hyper-NeRF (Park et al. (2021b)) and Neural 3D Video (N3DV) (Li et al. (2022)). **Hyper-NeRF** dataset was captured using a handheld rig equipped with two Pixel 3 cameras. We utilize data from one camera and conduct evaluations on the held-out views captured by the other. **N3DV** dataset consists of 18–20 synchronized cameras per scene, recording 10–30 second sequences. To conduct monocular experiments, we follow the experimental protocol of MoDGS (Qingming et al. (2025)), using cam0 for training and reporting evaluations on cam5 and cam6. For quantitative evaluation, we employ three standard metrics: Peak Signal-to-Noise Ratio (PSNR), Structural Similarity Index (SSIM) (Wang et al. (2004)), and Learned Perceptual Image Patch Similarity (LPIPS) (Zhang et al. (2018)).

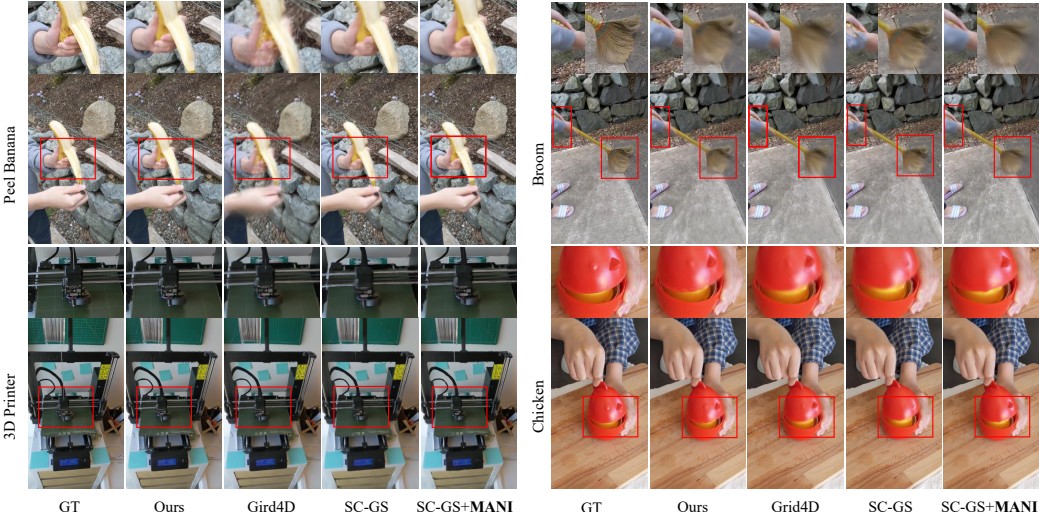

Figure 2. **Qualitative comparison** on the Hyper-NeRF(vrig) dataset (Park et al. (2021b)). Compared with other SOTA methods,our method reconstructs finer details of the moving objects.

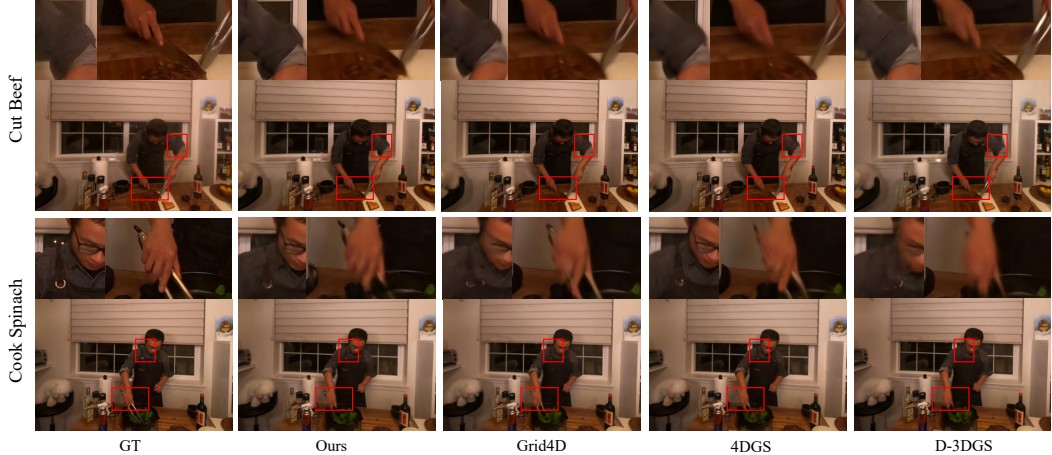

Figure 3. **Qualitative comparison** on the N3DV dataset (Li et al. (2022)).

**Baselines and Implementation.** We compare our method with state-of-the-art methods in dynamic scene reconstruction, including NeRF-based methods (TiNeuVox (Fang et al. (2022)), Hyper-NeRF (Park et al. (2021b)), HexPlanes (Cao & Johnson (2023))) and 3DGS-based methods (D-3DGS (Yang et al. (2024b)), 4DGS (Wu et al. (2024)), ED3DGS (Bae et al. (2024)),MoDec-GS (Kwak et al. (2025)), Grid4D (Xu et al. (2024)), SC-GS (Huang et al. (2023)), MoDGS (Qingming et al. (2025))). All implementations are based on PyTorch framework and trained on a single V100 GPU with 32 GB of VRAM. For more implementation details, please refer to Appendix A.2.

## 5.2 COMPARISONS

**Results on Hyper-NeRF.** As shown in Table 1, our method outperforms state-of-the-art baselines across all scenes and evaluation metrics. The qualitative results in Figure 2 further illustrate that our approach captures scene dynamics with higher fidelity, producing more complete and detailed reconstructions of moving objects. In addition, we augment SC-GS (Huang et al. (2023)) with our Motion-Adaptive Node Initialization (MANI), denoted as SC-GS+MANI. The last three rows of Table 1 show that SC-GS+MANI achieves clear improvements over the original SC-GS, and this advantage is also visible in Figure 2: for instance, in the Broom and Chicken scenes, SC-GS+MANI reconstructs dynamic regions more thoroughly with richer details, benefiting from the motion-aware initialization of control nodes. More results are available in Appendix A.4.

(a) Key components

| Method | PSNR↑ | SSIM↑ | LPIPS↓ |
|---|---|---|---|
| baseline | 22.35 | 0.613 | 0.335 |
| +MANI | 23.89 | 0.635 | 0.315 |
| +MS | 24.51 | 0.658 | 0.278 |
| +MS (w/o Init) | 24.13 | 0.639 | 0.284 |
| **Ours** | **25.78** | **0.722** | **0.242** |

(b) Node Init.

| Method | PSNR↑ | SSIM↑ | LPIPS↓ |
|---|---|---|---|
| FPS | 24.49 | 0.678 | 0.280 |
| Voxel | 24.06 | 0.652 | 0.271 |
| Tracklet | 24.83 | 0.681 | 0.253 |
| **MANI (ours)** | **25.78** | **0.722** | **0.242** |

(c) Node Traj.

| Method | PSNR↑ | SSIM↑ | LPIPS↓ |
|---|---|---|---|
| MLP | 23.95 | 0.633 | 0.317 |
| Grid | 24.28 | 0.649 | 0.271 |
| Tracklet | 24.59 | 0.671 | 0.263 |
| Linear | 23.15 | 0.590 | 0.384 |
| **MS (ours)** | **25.78** | **0.722** | **0.242** |

Table 3. **Ablation studies** on the Hyper-NeRF (Park et al. (2021b)) dataset.

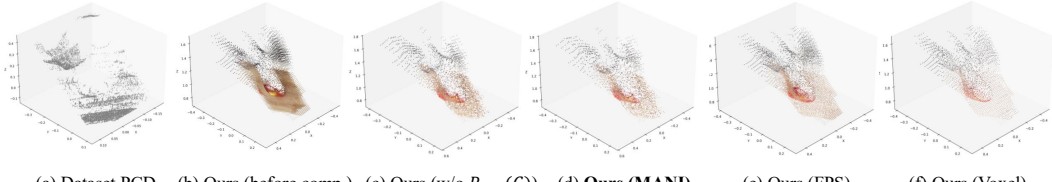

(a) Dataset PCD  (b) Ours (before comp.)  (c) Ours (w/o $P_{dyn}(C)$)  (d) **Ours (MANI)**  (e) Ours (FPS)  (f) Ours (Voxel)

Figure 4. **Visualization** of different Node init. meth. on Chicken scene of Hyper-NeRF data (Park et al. (2021b)).

**Results on N3DV.** Table 2 reports the per-scene results on the N3DV dataset. Under the monocular setting, our method achieves state-of-the-art performance with a mean PSNR of 23.31 dB. Figure 3 provides qualitative comparisons, where the highlighted red boxes show sharper and more coherent motion with fewer artifacts. For example, in fast hand motions, our method produces clearer contours and structures, while others yield blurry reconstructions. These improvements arise from placing more control points in motion-dominant areas and modeling their trajectories with spline parameterization, offering a robust alternative to implicit MLP deformation fields.

## 5.3 ABLATION STUDY

We conduct ablation studies on our method using the Hyper-NeRF (Park et al. (2021b)) dataset, and summarize the results in Table 3, Figure 4 and Figure 5. Our baseline follows a design similar to SC-GS (Huang et al. (2023)), with more details provided in Appendix A.3.

**Motion-Adaptive Node Initialization (MANI).** As shown in Table 3a, introducing MANI on top of the baseline yields clear performance gains. Table 3b further compares MANI with alternative initialization strategies (FPS (Huang et al. (2023)), voxel-based (Kong et al. (2025)), tracklet-based (Liang et al. (2025))), confirming the superiority of our motion-adaptive design. Figure 4 visualizes the initialization. (a) shows the raw point cloud provided by the dataset, where COLMAP (Schonberger & Frahm (2016)) fails to recover dynamic regions due to view inconsistency, causing static sampling to poorly cover moving areas.(b) shows our patch-to-node strategy yields better distribution, with red region indicating dynamic area in Chicken scene. (c,d) shows adding the dynamic tendency score $P_{dyn}(C)$ (Eq. 7) further merges static redundancy and preserves dynamic details. (e,f) shows replacing our strategy with FPS or voxel-based initialization results in inferior performance.

**Spline-Parameterized Node Trajectories (MS).** As shown in Table 3a, adding MS to the baseline (row 3) yields a significant performance gain, and initializing node splines with 2D tracklets from VFM models (row 4) further boosts the results. To validate its effectiveness, we replace MS with alternative deformation methods, including an MLP (Yang et al. (2024b)), a grid-based method (Wu et al. (2024)), and a tracklet-based method (Liang et al. (2025)). Table 3c reports the quanti-

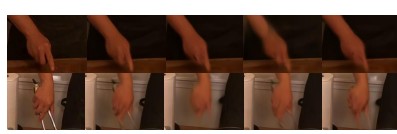

GT  **Ours**  MLP  Grid  Tracklet

Figure 5. Qualitative results of ablation.

tative results. MLP and grid-based approaches suffer from entangled optimization with large parameter spaces, leading to suboptimal performance under sparse control nodes. Tracklet-based deformation benefits from motion priors and achieves better reconstruction, but its reliance on predicted trajectories and clustering introduces noise, resulting in less stable optimization. In addition, qualitative results on the N3DV dataset (Figure 5) show that our method produces clearer and more complete reconstructions of dynamic regions.

## 6 CONCLUSION

In this work, we introduced a motion-adaptive framework for dynamic 3D Gaussian Splatting that addresses the imbalance between static redundancy and dynamic insufficiency in existing sparse control methods. By leveraging vision foundation model priors for node initialization, applying motion-aware compression to adapt representational density, and employing a spline-based trajectory formulation for stable optimization, our approach achieves substantial improvements in reconstruction quality. Extensive experiments validate its superiority over prior state-of-the-art methods, highlighting the effectiveness of aligning node allocation with motion complexity. Looking ahead, we believe this framework opens the door to incorporating stronger motion priors and handling more complex topological variations in dynamic scenes.

## 7 ACKNOWLEDGEMENTS

This work was in part supported by the Strategic Priority Research Program of the Chinese Academy of Sciences under Grant No. XDA0450203, and the National Natural Science Foundation of China under Grant 62172392, and the Innovation Research Program of ICT CAS (E261070).

## ETHICS STATEMENT

This work does not raise any ethical concerns. It does not involve human subjects, personally identifiable information, or sensitive data. No potentially harmful insights, methodologies, or applications are introduced. The datasets and models used are publicly available and widely adopted in prior research. We have complied with all relevant ethical standards and the ICLR Code of Ethics.

## REPRODUCIBILITY STATEMENT

We have made significant efforts to ensure the reproducibility of our work. The paper provides detailed descriptions of the proposed methodology in Section 4. Complete model architecture, training settings, experimental protocols, hyperparameters, and evaluation metrics are documented in Section 5 and Appendix A.2. All datasets used are publicly available and the preprocessing steps are described in the appendix.

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

# A APPENDIX

## A.1 LLM USAGE

In this work, large language models (LLMs) were only used as a general-purpose writing assist tool. Specifically, LLMs were employed for correcting grammatical errors and refining the language style of the manuscript. No part of the research ideation, methodology design, experiments, analysis, or results was generated by LLMs. The authors take full responsibility for the content of this paper.

## A.2 ADDITIONAL TRAINING DETAILS

**Loss functions and weights.** We provide a detailed explanation for each term of the loss in Eq. 13. We employ two categories of loss functions to supervise the learning of dynamic Gaussian primitives.

To ensure that rendered observations align with the input supervision signals, we impose per-frame reconstruction objectives on color, depth, and mask predictions. At each training iteration, given the camera parameters, we render an image $\hat{I}_t$, a depth map $\hat{D}_t$, and a mask $\hat{M}_t$ following Eq. 1. The RGB loss $\mathcal{L}_{\text{rgb}}$ consists of a weighted combination of mean squared error (MSE) between $\hat{I}_t$ and $I_t$ (weight 0.8), and a D-SSIM loss (Wang et al. (2004)) (weight 0.2). The depth loss $\mathcal{L}_{\text{depth}}$ computes the MSE between $\hat{D}_t$ and the monocular depth prediction $D_t$ (Hu et al. (2025)), with a weight of 1.0. The mask loss $\mathcal{L}_{\text{mask}}$ enforces consistency between $\hat{M}_t$ and the foreground mask $M_t$ predicted by a segmentation model (Yang et al. (2023)), also weighted by 1.0.

To regularize temporal correspondences and guide the motion of Gaussians, we introduce a tracking loss $\mathcal{L}_{\text{track}}$, composed of a 2D trajectory term $\mathcal{L}_{\text{track}\rightarrow 2d}$ and a depth reprojection term $\mathcal{L}_{\text{track}\rightarrow depth}$. For randomly sampled query time $t$ and target time $t'$, the 2D trajectory loss $\mathcal{L}_{\text{track}\rightarrow 2d}$ measures the MSE between the rendered trajectory $\hat{u}_{t\rightarrow t'}$ and the tracked trajectory $u_{t\rightarrow t'}$ provided by a pretrained tracker (Doersch et al. (2023)), under normalized pixel coordinates, with a weight of 2.0. Meanwhile, the depth reprojection loss $\mathcal{L}_{\text{track}\rightarrow depth}$ penalizes the discrepancy between the rendered reprojection depth $\hat{d}_{t\rightarrow t'}$ and the metric-aligned depth $\hat{D}(u_{t\rightarrow t'})$ (Hu et al. (2025)), with a weight of 0.1, ensuring physically plausible motion supervision.

To further constrain the motion of Gaussians and avoid degenerate deformations, we adopt an as-rigid-as-possible (ARAP) loss $\mathcal{L}_{\text{arap}}$ inspired by physics-based shape regularization (Huang et al. (2024); Lei et al. (2025)). Specifically, given two timesteps $t$ and $t'$ separated by a fixed interval $\Delta$, the loss is formulated as

$$\mathcal{L}_{\text{arap}} = \sum_{t=1}^{T}\sum_{j=1}^{N_g}\sum_{k\in\hat{\mathcal{E}}(j)} \lambda_l \left| \|\mathbf{p}_t^{(j)} - \mathbf{p}_t^{(k)}\| - \|\mathbf{p}_{t'}^{(j)} - \mathbf{p}_{t'}^{(k)}\| \right|$$
$$+ \lambda_c \left\| \mathbf{Q}_t^{(k)-1}\mathbf{p}_t^{(j)} - \mathbf{Q}_{t'}^{(k)-1}\mathbf{p}_{t'}^{(j)} \right\|, \tag{14}$$

where $\hat{\mathcal{E}}(j)$ denotes the neighborhood of Gaussian $\mathcal{G}_j$, $\mathbf{p}_t^{(j)}$ is the 3D position of $\mathcal{G}_j$ at time $t$, and $\mathbf{Q}_t^{(k)}$ is the local frame constructed around $\mathcal{G}_k$. The first term encourages the pairwise distances between neighboring Gaussians to remain stable across timesteps, while the second term preserves the relative local coordinates under the corresponding local frames.

**Dataset details.** **Hyper-NeRF** (Park et al. (2021b)) provides dynamic scenes with continuous viewpoints, where each timestamp exhibits complex topological deformations. We adopt four scenes from this dataset, training and rendering at a resolution of $960 \times 640$. In our setting, we employ the "vrig" subset, which was recorded with stereo cameras, using one camera's sequence for training and the other for validation. **Neural 3D Video** (Li et al. (2022)) contains 15–20 multi-view videos, each consisting of 300 frames. Total six scenes are used to train and render at a resolution of $1352 \times 1014$. For the Flame Salmon scene, we utilize the initial 300 frames from its 1200-frame sequence in our experiments. Following the experimental protocol of MoDGS (Qingming et al. (2025)), we use

Table 4. Additional ablation study on different Depth prior on Chiken scene of Hyper-NeRF dataset per-scene.

| Method | PSNR↑ | SSIM↑ | LPIPS↓ |
|---|---|---|---|
| DepthCrafter Hu et al. (2025) | 29.61 | 0.865 | 0.165 |
| DepthAnything Yang et al. (2024a) | 29.43 | 0.848 | 0.164 |
| Metric3D Yin & Hu (2024) | 29.38 | 0.855 | 0.173 |

Table 5. Additional ablation study on different 2D Tracklets prior on Chiken scene of Hyper-NeRF dataset per-scene.

| Method | PSNR↑ | SSIM↑ | LPIPS↓ |
|---|---|---|---|
| TAPIR Doersch et al. (2023) | 29.66 | 0.863 | 0.161 |
| CoTracker Karaev et al. (2024) | 29.20 | 0.839 | 0.182 |
| SpatialTracker Xiao et al. (2024) | 29.47 | 0.857 | 0.169 |

cam0 for training and report evaluations on cam5 and cam6. We generate initial point clouds for each scene following 4DGS (Wu et al. (2024)).

**Implementation details.** We use Adam (Kingma (2014)) to optimize our method and Gaussians in canonical space jointly. We fine-tune our optimization parameters by the configuration outlined in the 3DGS (Kerbl et al. (2023)). Besides, the adaptive density control of Gaussians from original 3DGS is also applied. The learning rates of mean, scale, rotation, opacity and color of Gaussian are set to $1.6 \times 10^{-4}, 5 \times 10^{-3}, 1 \times 10^{-3}, 1 \times 10^{-2}$ and $1 \times 10^{-2}$, respectively.

We extract per-patch token embeddings for a set of keyframe images $\{I_t\}$ using VGGT (Wang et al. (2025)). The model alternates frame-wise and global self-attention layers, enabling the tokens to encode both intra-frame semantics and inter-frame correspondence across all views. Consequently, the resulting representations accumulate temporal context: tokens in stationary background regions remain highly consistent across frames (higher inter-frame cosine similarity), whereas tokens associated with moving foreground objects vary more strongly (lower similarity). We exploit this property by computing cross-frame token similarities to obtain motion-aware cues that later guide initialization and regularization. To supply complementary priors and supervision signals, we adopt off-the-shelf vision foundation models: Track-Anything (Yang et al. (2023)) for foreground segmentation masks, DepthCrafter (Hu et al. (2025)) for temporally consistent monocular video depth, and TAPIR (Doersch et al. (2023)) for dense 2D point trajectories. These components provide object masks, long-range consistent depth sequences, and per-point tracks, respectively, which we integrate into the training objectives and the construction of motion-aware priors.

### A.3 ADDITIONAL ABLATIONS

**Details of ablation setting.** Our baseline implementation follows a design similar to SC-GS (Huang et al. (2023)), where Nodes are sampled from the input point cloud using farthest point sampling (FPS) and their motions are parameterized by an MLP. To model Gaussian dynamics, we replace the conventional linear blend skinning (LBS) (Sumner et al. (2007)) with deformation via quaternion-based blending (DQB) (Kavan et al. (2007)), which serves as the backbone deformation mechanism in all ablation settings.

**Additional ablation study on VFM prior.** We further evaluate the impact of different VFM-based depth estimation models on the Hyper-NeRF dataset, using the Chicken scene as a representative case. Specifically, we compare DepthCrafter (Hu et al. (2025)), DepthAnything (Yang et al. (2024a)), and Metric3D (Yin & Hu (2024)), as summarized in Table 4. The results indicate that DepthCrafter provides relatively more reliable results in our setting. Therefore, we adopt DepthCrafter as the depth prior in our framework. We further evaluate different 2D tracklet models on the same Chicken scene of the Hyper-NeRF dataset, comparing TAPIR (Doersch et al. (2023)), CoTracker (Karaev et al. (2024)), and SpatialTracker (Xiao et al. (2024)), as reported in Table 5. TAPIR integrates more smoothly into our pipeline and yields more reliable tracklets under the dynamic scenes we consider. Consequently, we employ TAPIR as our default tracking module.

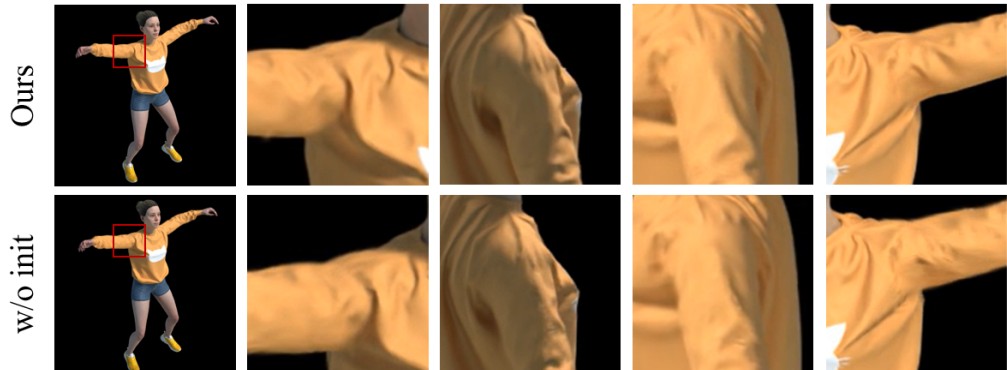

Figure 6. Additional ablation study on rotational trajectory initialization on JumpingJacks scene in the D-NeRF dataset (Pumarola et al. (2021)).

Table 6. Ablation study on VFM prior loss.

| Method | PSNR↑ | SSIM↑ | LPIPS↓ |
|---|---|---|---|
| w/o $L_{mask}$ | 25.46 | 0.691 | 0.259 |
| w/o $L_{depth}$ | 24.97 | 0.674 | 0.277 |
| w/o $L_{track}$ | 25.52 | 0.690 | 0.253 |
| **Ours** | **25.78** | **0.722** | **0.242** |

**Additional ablation study on rotational trajectory initialization.** Hyper-NeRF (Park et al. (2021b)) already includes several scenes with noticeable rotational motion, such as Banana and Chicken, where our trajectory initialization is applied without any modification. To further isolate articulated rotations, we additionally evaluate the proposed method on a D-NeRF (Pumarola et al. (2021)) scene (JumpingJacks) that exhibits strong joint rotation, using exactly the same loss weights as in the main experiments. As illustrated in Figure 6, the variant with trajectory initialization consistently produces sharper reconstructions and improved quantitative metrics, while the model trained without initialization still converges to a reasonable solution. We do not observe slower convergence or the need to increase regularization strength in any of these settings, which suggests that the proposed initialization remains stable and effective on rotational motion.

**Additional ablation study on VFM prior loss.** To assess the sensitivity to VFM based supervision, we ablate the loss terms associated with masks, depth and tracking. As shown in Table 6, removing each term leads to only moderate drops in PSNR, SSIM and LPIPS, and the optimization remains stable. These results indicate that the framework does not rely on any single VFM prior. When the depth and tracking cues are weakened, the system effectively behaves as a sparse control dynamic 3DGS that is mainly driven by RGB reconstruction and geometric regularizers rather than being dominated by potentially erroneous VFM signals.

**Additional ablation study on hyperparameters.** The framework involves several hyperparameters, but many of them are either learned or defined in an adaptive way. The RBF radius is optimized jointly with other parameters, and the per voxel compression ratio $r\%(C)$ is computed from the dynamic tendency of each voxel, so neither requires manual tuning. The MANI weights $(\alpha, \beta, \eta)$ only balance semantic similarity and foreground priors and are kept fixed for all scenes. The compression bounds $(r_{min}, r_{max})$ and the spline keyframe interval $N$ are selected once on a validation scene and reused in all experiments. Table **??** report ablations on $(r_{min}, r_{max})$, $\eta$, $(\alpha, \beta)$ and $N$ on the Chicken and 3D-Printer scenes from Hyper-NeRF. Performance varies smoothly within a broad range of values and the default configuration lies close to the optimum, indicating that the method is not overly sensitive and does not require per-scene retuning.

Table 7. Ablation study on $(r_{min}, r_{max})$ combinations.

| Method | PSNR↑ | SSIM↑ | LPIPS↓ |
|---|---|---|---|
| $[25, 50]$ | 25.87 | 0.774 | 0.199 |
| $[25, 75]$ | 26.10 | 0.796 | 0.196 |
| $[50, 75]$ | 26.03 | 0.790 | 0.196 |

Table 8. Ablation study on $\eta$.

| Method | PSNR↑ | SSIM↑ | LPIPS↓ |
|---|---|---|---|
| 0 | 25.15 | 0.713 | 0.236 |
| 0.25 | 25.81 | 0.768 | 0.215 |
| 0.5 | 26.10 | 0.796 | 0.196 |
| 0.75 | 25.95 | 0.785 | 0.209 |

## A.4 ADDITIONAL RESULTS

**Efficiency comparison.** We provide detailed quantitative results on the Hyper-NeRF dataset in Table 11, reporting per-scene PSNR, training time, and rendering speed (FPS). Furthermore, Table 12 presents a comprehensive comparison of our method against representative NeRF-based and 3DGS-based approaches, including PSNR, training time, rendering speed (FPS), and storage size (MB) at a resolution of $536 \times 960$. Specifically, the results of Nerfies, HyperNeRF, TiNeuVox-B, D-3DGS, and 4DGS are taken from (Wu et al. (2024)), measured on an NVIDIA RTX 3090 GPU, while MoDec-GS is reported in (Kwak et al. (2025)) using an RTX A6000 GPU. Our method is evaluated on an NVIDIA V100 GPU. Due to time constraints, we have not yet conducted performance benchmarking on the same hardware. Nevertheless, it is well established that the V100 provides lower computational throughput than both RTX 3090 and RTX A6000. Therefore, the favorable comparison results demonstrate the inherent efficiency and effectiveness of our approach despite the hardware disadvantage.

**Additional quantitative comparison.**

To further validate the effectiveness of our approach, we conduct additional experiments on the N3DV (Li et al. (2022)) dataset under the multi-view setting and compare our method against several state-of-the-art baselines, as reported in Table 13. The results demonstrate that our method also achieves strong performance in the multi-view scenario. To further assess performance under high motion scenarios, we additionally evaluate on the Nvidia dataset (Yoon et al. (2020)). As reported in Table 14, the proposed method achieves the best mean PSNR across all four sequences and attains the best or second best PSNR on each individual scene, outperforming representative dynamic 3D Gaussian baselines.

**Additional qualitative comparison.** We conduct additional qualitative comparison on the Hyper-NeRF dataset (Park et al. (2021b)), comparing our method with 4DGS (Wu et al. (2024)), Grid4D (Xu et al. (2024)), and D-3DGS (Yang et al. (2024b)), as shown in Figure 7 and Figure 8.

Table 9. Ablation study on $(\alpha, \beta)$.

| Method | PSNR↑ | SSIM↑ | LPIPS↓ |
|---|---|---|---|
| $[0.25, 0.75]$ | 25.86 | 0.778 | 0.199 |
| $[0.5, 0.5]$ | 26.05 | 0.790 | 0.197 |
| $[0.75, 0.25]$ | 26.10 | 0.796 | 0.196 |
| $[1, 0]$ | 26.03 | 0.788 | 0.197 |

Table 10. Ablation study on keyframe interval $N$.

| Method | PSNR↑ | SSIM↑ | LPIPS↓ | FPS↑ |
|---|---|---|---|---|
| 2 | 25.94 | 0.782 | 0.215 | 101 |
| 6 | 26.14 | 0.814 | 0.193 | 97 |
| 8 | 26.10 | 0.796 | 0.196 | 90 |
| 12 | 25.88 | 0.776 | 0.224 | 86 |

Table 11. The training times and rendering speed on Hyper-NeRF dataset per-scene.

| Scene | Broom | 3D-Printer | Chicken | Banana | Mean |
|---|---|---|---|---|---|
| PSNR↑ | 22.37 | 22.53 | 29.66 | 28.55 | 25.78 |
| Training Times (m) ↓ | 48 | 37 | 30 | 41 | 39 |
| FPS↑ | 61 | 92 | 88 | 37 | 69.5 |

Table 12. **Efficiency comparison** on Hyper-NeRF dataset. We highlight the best , second best and the third best results in each scene.

| Methods | PSNR↑ | Training Times↓ | FPS↑ | Storage(MB)↓ |
|---|---|---|---|---|
| Nerfies Park et al. (2021a) | 22.2 | h | <1 | - |
| HyperNeRF Park et al. (2021b) | 22.4 | 32h | <1 | - |
| TiNeuVox-B Fang et al. (2022) | 24.3 | 30m | 1 | 48 |
| D-3DGS Yang et al. (2024b) | 19.7 | 40m | 55 | 52 |
| 4DGS Wu et al. (2024) | 25.2 | 30m | 34 | 61 |
| MoDec-GS Kwak et al. (2025) | 25 | 1.2h | 23.8 | 28 |
| **Ours** | 25.8 | 39m | 70 | 25 |

Table 13. **Additional quantitative comparison** on N3DV dataset per-scene. We highlight the best , second best and the third best results in each scene.

| Method | Coffee Martini | | Cook Spinach | | Cut Beef | | Flame Salmon | | Flame Steak | | Sear Steak | | Mean | |
|---|---|---|---|---|---|---|---|---|---|---|---|---|---|---|
| | PSNR↑ | SSIM↑ | PSNR↑ | SSIM↑ | PSNR↑ | SSIM↑ | PSNR↑ | SSIM↑ | PSNR↑ | SSIM↑ | PSNR↑ | SSIM↑ | PSNR↑ | SSIM↑ |
| K-Planes Fridovich-Keil et al. (2023a) | 29.99 | 0.943 | 32.60 | 0.968 | 31.82 | 0.965 | 30.44 | 0.942 | 32.39 | 0.970 | 32.52 | 0.971 | 31.63 | 0.960 |
| HyperReel Attal et al. (2023) | 28.37 | 0.892 | 32.30 | 0.941 | 32.92 | 0.945 | 28.26 | 0.882 | 32.20 | 0.949 | 32.57 | 0.952 | 31.10 | 0.927 |
| 4DGS Wu et al. (2024) | 28.39 | 0.944 | 32.61 | 0.971 | 32.07 | 0.966 | 29.14 | 0.948 | 33.43 | 0.977 | 32.85 | 0.977 | 31.42 | 0.964 |
| E-3DGS Bae et al. (2024) | 29.10 | 0.947 | 32.95 | 0.957 | 33.56 | 0.970 | 29.61 | 0.949 | 33.57 | 0.974 | 33.45 | 0.974 | 32.04 | 0.962 |
| Grid4D Xu et al. (2024) | 28.34 | 0.938 | 32.44 | 0.971 | 33.23 | 0.974 | 28.89 | 0.947 | 32.20 | 0.980 | 33.15 | 0.978 | 31.38 | 0.965 |
| **Ours** | 29.21 | 0.950 | 32.95 | 0.968 | 33.91 | 0.981 | 30.53 | 0.954 | 33.87 | 0.982 | 33.82 | 0.983 | 32.38 | 0.970 |

Table 14. **Additional quantitative comparison** on Nvidia (Yoon et al. (2020)) dataset.

| Method | Balloon1 | Balloon2 | Jumping | Umbrella | Mean |
|---|---|---|---|---|---|
| Deformable 3DGS Yang et al. (2024b) | 15.91 | 15.13 | 16.68 | 17.26 | 16.25 |
| 4DGS Wu et al. (2024) | 21.89 | 24.85 | 22.37 | 22.36 | 22.87 |
| HiMoR Liang et al. (2025) | 23.90 | 23.48 | 20.04 | 24.30 | 22.93 |
| MoSca | 23.58 | 27.80 | 25.01 | 25.17 | 25.39 |
| Ours | 24.39 | 27.65 | 25.43 | 25.69 | 25.79 |

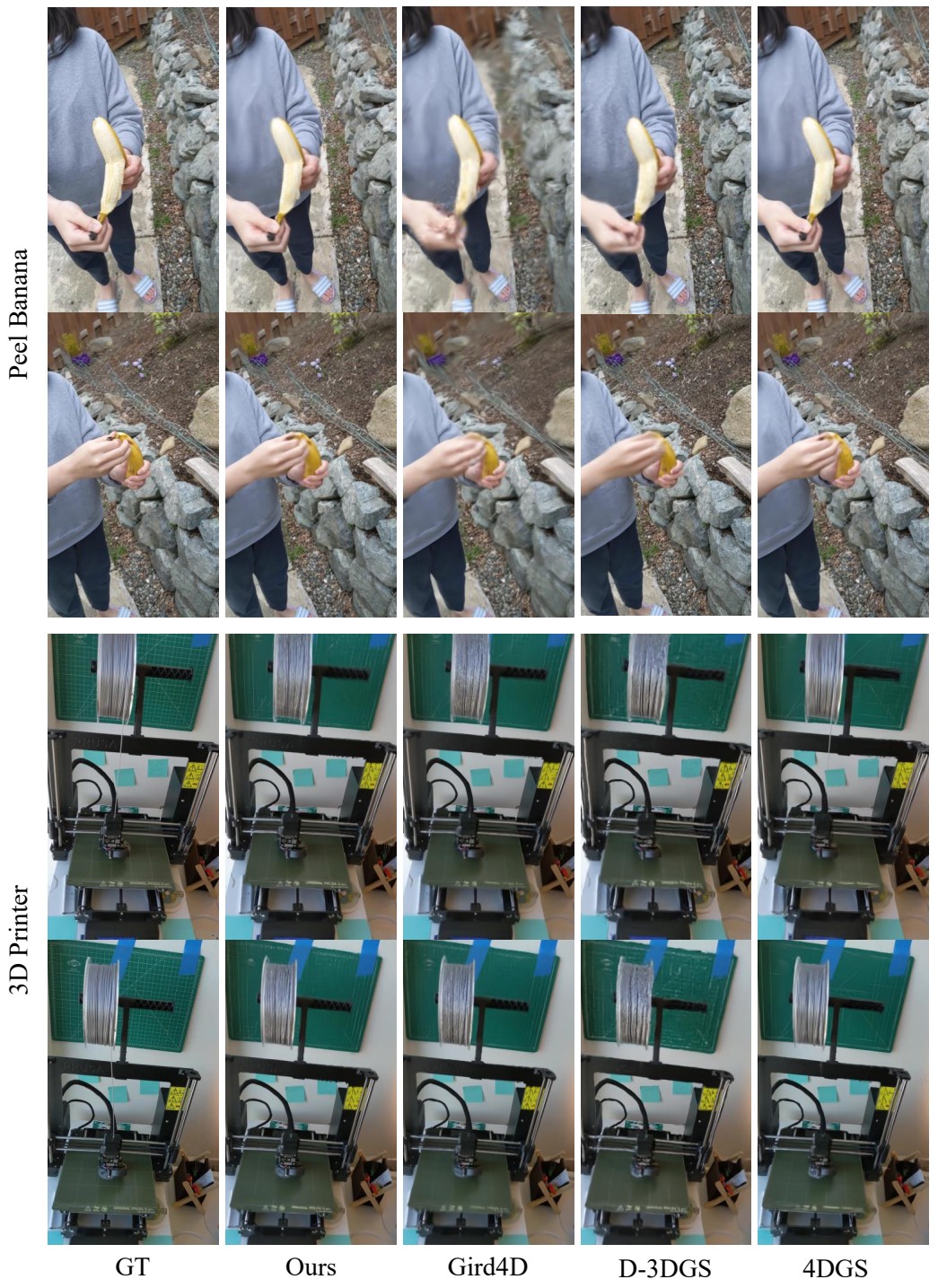

Figure 7. **Additional qualitative comparison** on Peel Banana and 3D Printer scene in the HyperNeRF dataset (Park et al. (2021b)).

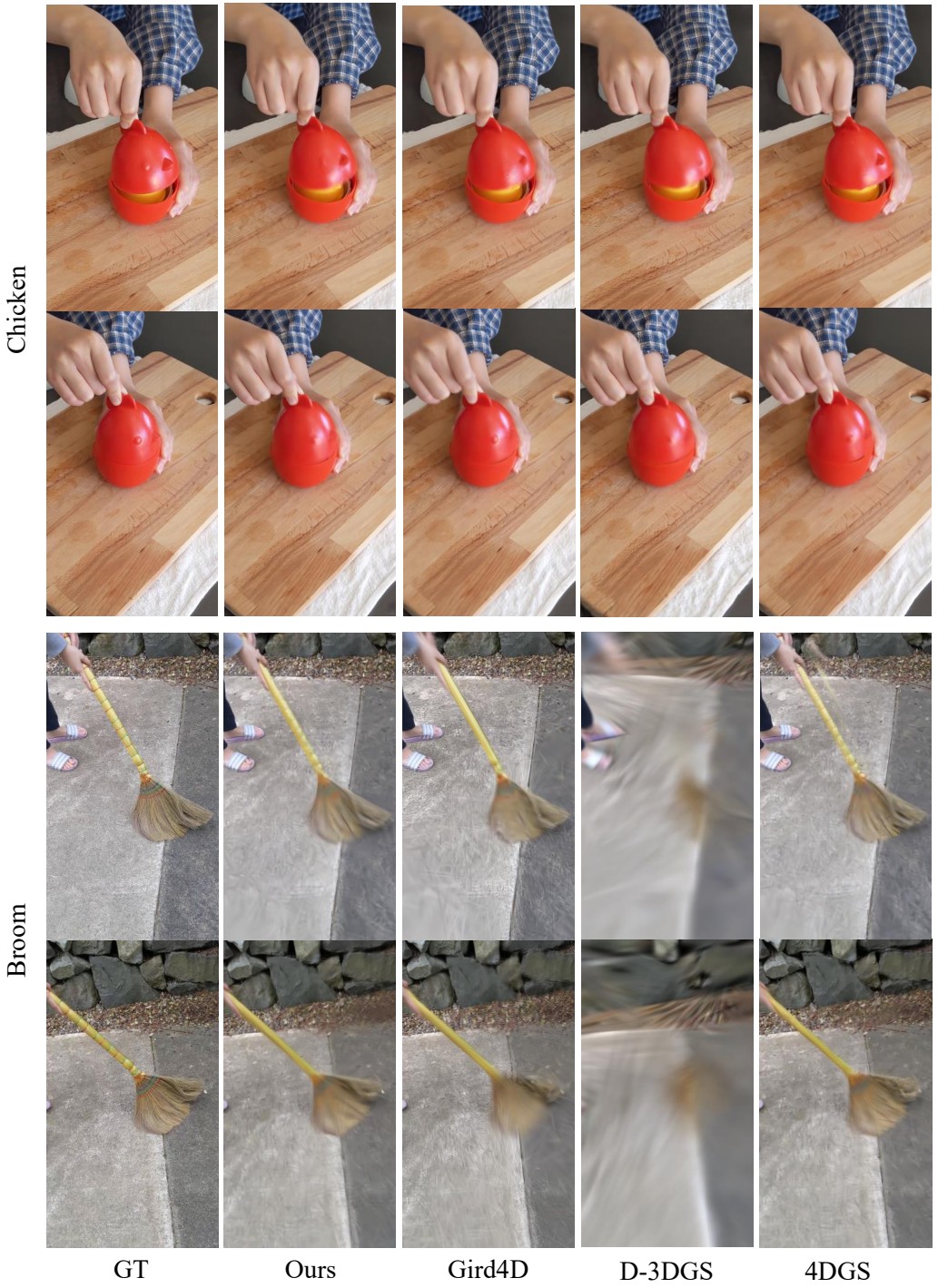

Figure 8. **Additional qualitative comparison** on Chicken and Broom scene in the HyperNeRF dataset (Park et al. (2021b)).

