# OpenReview forum: "From Tokens to Nodes: Semantic-Guided Motion Control for Dynamic 3D Gaussian Splatting"
_ICLR.cc/2026/Conference — ICLR 2026 Poster_

### Official Review · Reviewer_JNHZ · 2025-10-28

**Soundness:** 3
**Presentation:** 2
**Contribution:** 1
**Rating:** 4
**Confidence:** 4

**Summary:**

This paper proposes a semantic-guided motion-adaptive framework for dynamic 3D Gaussian Splatting (3DGS) reconstruction from monocular videos. The key idea is to use vision foundation models to extract semantic and motion priors that guide node allocation (“From Tokens to Nodes”) and introduce spline-based trajectory parameterization for node motion. The authors claim that this design aligns control-point density with motion complexity and yields smoother, more stable optimization. Experiments on Hyper-NeRF and N3DV datasets are presented, showing improved PSNR/SSIM compared to SC-GS, Grid4D, and other baselines.

However, after close inspection, the technical novelty appears limited. Several core components replicate or closely follow prior works such as SC-GS, Superpoint-GS, H3D-DGS, and MoSca. The improvements demonstrated experimentally are modest and insufficiently supported by analysis or ablation. The paper is well written, but the contributions are overstated relative to their actual methodological advancement.

**Strengths:**

1. **Clarity and organization** – The paper is clearly structured, with explicit method description, and comprehensive visuals (Figure 1–3). The pipeline is easy to follow.
2. **Integration of semantic priors** – The idea of leveraging pretrained vision foundation models (VFM) for adaptive node placement is conceptually appealing and fits the current trend of combining large-scale priors with geometric reconstruction.
3. **Empirical validation** – The paper reports quantitative and qualitative results on two widely used dynamic scene datasets (Hyper-NeRF, N3DV), demonstrating that the proposed pipeline functions in practice.

**Weaknesses:**

1. **Lack of genuine novelty** –
   - The *node-based deformation representation* (Sec. 4.2) is almost identical to prior works such as SC-GS [1], Superpoint-GS [2], and H3D-DGS [3]. The node weighting mechanism and dual quaternion interpolation are directly inherited from SC-GS and MoSca[4] with minimal modification.
   - The *image-to-space projection* in Sec. 4.3 follows the same design as H3D-DGS, where control points are back-projected from image patches using estimated depth. The only difference is the additional compression heuristic, which is conceptually similar to spatial clustering used in K-Means-based approaches (e.g., H3D-DGS achieving similar sparsity with 10% of control points).

2. **Experimental insufficiency** –
   - Key baselines such as 4DGS [5] are missing in several figures (e.g., Figure 2), and qualitative results appear cherry-picked. In Figure 3, for example, the Cook Spinach sequence shows that 4DGS visually outperforms the proposed method in both facial and object regions.
   - The claimed superiority is not consistently demonstrated: PSNR/SSIM gains over strong baselines are marginal (≤1 dB) and may not be statistically significant.
   - No comparison against simpler node initialization (e.g., K-Means clustering GS according to spatial position) or trajectory models (linear vs. spline) is provided to justify additional complexity.
   - The *spline-based trajectory* (Sec. 4.4) merely replaces existing linear or MLP deformation with cubic interpolation. Given that both N3DV [6] and Hyper-NeRF [7] are low-motion datasets, linear interpolation would likely suffice. No temporal interpolation or high-motion sequence evaluation is provided.

```
[1] Huang, Yi-Hua, et al. "Sc-gs: Sparse-controlled gaussian splatting for editable dynamic scenes." Proceedings of the IEEE/CVF conference on computer vision and pattern recognition. 2024.

[2] Wan, Diwen, Ruijie Lu, and Gang Zeng. "Superpoint gaussian splatting for real-time high-fidelity dynamic scene reconstruction." arXiv preprint arXiv:2406.03697 (2024).

[3] He, Bing, et al. "S4d: Streaming 4d real-world reconstruction with gaussians and 3d control points." arXiv preprint arXiv:2408.13036 (2024).

[4] Lei, Jiahui, et al. "Mosca: Dynamic gaussian fusion from casual videos via 4d motion scaffolds." Proceedings of the Computer Vision and Pattern Recognition Conference. 2025.

[5] Wu, Guanjun, et al. "4d gaussian splatting for real-time dynamic scene rendering." Proceedings of the IEEE/CVF conference on computer vision and pattern recognition. 2024.

[6] Li, Tianye, et al. "Neural 3d video synthesis from multi-view video." Proceedings of the IEEE/CVF conference on computer vision and pattern recognition. 2022.

[7] Park, Keunhong, et al. "Hypernerf: A higher-dimensional representation for topologically varying neural radiance fields." arXiv preprint arXiv:2106.13228 (2021).
```

**Questions:**

See weaknesses.

---

> ### Author Response · Authors · 2025-11-28
>
> ### [W1&W2] Relation to Control-Point Deformation and H3D-DGS
>
> We appreciate the reviewer’s comments. To better clarify the positioning of our method, we **first** summarize its differences from SC-GS, SP-GS, MoSca, and H3D-DGS. In the original introduction, we already discussed SC-GS and MoSca, and we **have expanded the discussion** of  SP-GS and H3D-DGS in the revised one so that the contributions of our approach are stated more clearly.
>
> * **Node-based deformation representation.** As stated in the introduction, our work targets the mismatch between geometric uniformity and motion complexity in dynamic 3DGS, which leads to **static redundancy and dynamic insufficiency** when control points are allocated purely by geometry. In this setting, our deformation backbone intentionally follows prior control-point methods such as SC-GS and SP-GS, using distance based node weighting and dual quaternion blending. We do not claim novelty in this generic mixing scheme. The contribution is how control points are selected and how their motion is modeled: MANI uses VFM semantic tokens, foreground masks and motion tendency scores to allocate a sparse node set that reduces redundancy in static regions and increases coverage in dynamic regions, and MS parameterizes node trajectories as cubic splines. As shown in *Table 3*, adding a generic control-point backbone without MANI and MS brings only limited gains, while the full MANI plus MS configuration significantly improves over the baseline, which supports that the benefit comes from motion adaptive allocation and trajectory modeling rather than from reusing the deformation formulation itself.
>
> * **Image-to-space projection and relation to H3D-DGS**. Our image-to-space projection also uses depth and camera poses to back project image patches into 3D, in line with the design adopted by H3D-DGS. We follow this strategy as a standard and reliable way to lift 2D evidence into 3D candidates, and do not position it as a source of novelty. Building on this common step, the two frameworks instantiate different control structures and modeling goals. H3D-DGS introduces per-view ray frame control points whose motion is partly derived from optical flow, and is particularly well suited for streaming reconstruction with structured physical priors. In our setting, we instead construct a single global 3D node set that serves as motion anchors over time. Node locations are determined jointly by semantic tokens and motion tendency scores, and MANI compresses the candidate set so that a fixed node budget is concentrated on motion intensive and semantically important regions, directly addressing the static dynamic allocation issue discussed in the introduction. MS then learns spline trajectories in world space on top of this node set. Under comparable sparsity, our ablation experiments show that simple spatial clustering baselines such as FPS based or voxel based initialization perform consistently worse than MANI, suggesting that semantics and motion guided allocation provide additional benefits on top of purely geometric clustering.
>
> ------
>
> ### [W3] Clarification on Qualitative Results and 4DGS Baseline
>
> We appreciate the reviewer’s comments on the qualitative comparisons and 4DGS baselines. Regarding 4DGS on the HyperNeRF dataset, **its qualitative results are already included in the original appendix (Figures 7 and 8)**. In the main text, we only visualized SC-GS and SC-GS+MANI due to space limitations and because they more directly isolate the effect of our proposed MANI module within the same sparse control framework.
>
> For the Cook Spinach sequence in Figure 3, our goal is to show representative frames rather than to suggest that our method is visually superior in every region. Since the quantitative metrics are computed over entire sequences, it is natural that different methods may perform better in different local areas of some frames. Under the monocular setup, 4DGS produces slightly sharper facial details in certain frames, while our method provides clearer reconstructions for the hands and manipulated objects and achieves better sequence level quantitative performance overall. To provide a more complete picture and reduce any impression of cherry picking, **we have included rendered videos of Cook Spinach for both 4DGS and our method in the supplementary material**. We hope these additional results will assist the reviewer in assessing the temporal behavior of each method.

---

> > ### Comment · Reviewer_JNHZ · 2025-11-28
> >
> > thanks for your reply. For me, the paper has two major issues. The method's performance is not significant, and the experimental validation of the main contributions is insufficient.
> >
> > First, as you have replied to W3, the proposed method cannot consistently beat the 4DGS method. I have checked the provided Cook Spinach sequences reconstructed using both methods, and the reconstruction result is not visually better. From my perspective, qualitative results should support quantitative results, and claiming that achieving better quantitative results alone is convincing is not sufficient. 4DGS is a robust and efficient method that supports 4D reconstruction under both monocular and multi-camera settings. In contrast, control point methods usually suffer from slow training and convergence issues. If the visual quality is not consistently better, why bother designing such a complicated pipeline?
> >
> > Second, the main contributions are the MANI and MS modules. Even though you claimed in response to W1 & W2 that the effectiveness of these techniques is reflected in the final rendering results, I argue that the results are still not satisfactory. I insist that you should visualize both the distribution of control points in the scene and the motion trajectory of the control points. Figure 4 is extremely ambiguous, to the point that I cannot tell where these nodes correspond within the 3DGS representation, and I can hardly see how semantic tokens influence the distribution of control points. I recommend focusing on the main part of the dynamic scene and rendering Gaussians together with the control points. Taking the Cook Spinach sequence as an example, you can focus on the movement of the cooking man and visualize the control points correspondingly.

---

> > > ### Author Response · Authors · 2025-12-04
> > >
> > > We appreciate the reviewer’s insightful discussion. Regarding the concerns about performance and experimental validation, we would like to clarify three points.
> > >
> > > First, sparse-control methods are attractive because they greatly improve motion modeling efficiency compared to per-Gaussian prediction, but existing sparse-control approaches still suffer from unstable motion learning, with too few control nodes in dynamic regions and redundant nodes in static regions. Our method is explicitly designed to address this mismatch, and as a result it achieves both more efficient rendering (about **+27% FPS** on HyperNeRF, Table 12) and better reconstruction quality (**+0.32 dB** on HyperNeRF and **+0.68 dB** on N3DV, Tables 1 and 2). We would like to emphasize that by resolving this key issue in sparse-control representations, our method improves rendering efficiency and reconstruction quality at the same time, which we believe constitutes a substantial contribution. We therefore kindly ask the reviewer to consider not only PSNR-type image quality metrics, but also the strong rendering efficiency gains when assessing the overall impact of the method.
> > >
> > > Second, we agree that qualitative results should support quantitative metrics. At the same time, we would like to clarify that the reported metrics are statistics over all frames, and that the comparison videos provided in the supplementary material are consistent with these numbers. Concerning Figure 3 specifically, the representative frames shown there already indicate that our method better reconstructs regions with large motion (such as the moving hands in Cook Spinach), where the hand region is sharper and more stable than 4DGS.
> > >
> > > Third, regarding the contributions of the MANI and MS modules, the original submission already contains extensive ablations to isolate their effects. Beyond the standard module ablations in Table 3(a), we compare MANI against three common control-point initialization strategies in Table 3(b) and Figure 4, and compare MS against four alternative trajectory parameterizations in Table 3(c) and Figure 5, as well as analyzing the impact of different depth and 2D track VFMs in Tables 4 and 5. In the rebuttal, we further add ablations on three loss terms and six hyperparameters, together with additional qualitative visualizations for MS in Figure 6. We will also include the trajectory visualization rendering suggested by the reviewer in the final version. While we take the reviewer’s concerns seriously, we also note that reviewer `bxh3` finds that we "provide enough ablations to isolate the framework’s contributions", and reviewer `BzWF` comments that "the ablation studies effectively validate these contributions".

---

### Official Review · Reviewer_BzWF · 2025-10-29

**Soundness:** 3
**Presentation:** 3
**Contribution:** 3
**Rating:** 6
**Confidence:** 4

**Summary:**

This paper addresses the problem of dynamic 3D reconstruction from monocular videos using 3D Gaussian Splatting. It identifies a key limitation in current sparse control methods: control points are often allocated based purely on geometry (e.g., uniform sampling), leading to redundancy in static regions and insufficient density in dynamic regions. To overcome this, the paper proposes a motion-adaptive framework. The core contributions are: (1) Motion-Adaptive Node Initialization (MANI), which leverages semantic and motion priors from Vision Foundation Models (VFMs) via patch-token correspondences to compress nodes adaptively, concentrating control density in dynamic areas (2) Spline-Parameterized Node Trajectories (MS), which replaces MLP-based deformation fields with cubic splines initialized by 2D tracklets, aiming for smoother motion and more stable optimization. Experiments on the Hyper-NeRF and N3DV datasets demonstrate improvements in reconstruction quality and efficiency over state-of-the-art methods.

**Strengths:**

1.The paper accurately identifies and targets a fundamental limitation in existing sparse control methods for dynamic 3DGS – the mismatch between geometrically uniform control point allocation and non-uniform motion complexity. The concepts of static redundancy and dynamic insufficiency are well-articulated.
2.The proposed MANI method is highly novel and intuitive. Using VFM tokens to guide an adaptive compression process that preserves nodes in dynamic regions while aggressively merging them in static ones is a strong contribution. The visualization in Figure 4 effectively demonstrates this capability.
3.The method achieves state-of-the-art results on the Hyper-NeRF and N3DV datasets under the challenging monocular setting, outperforming numerous recent baselines quantitatively and qualitatively. The ablation studies effectively validate the contributions.

**Weaknesses:**

1.The proposed framework introduces a significant number of hyperparameters. These include parameters for node binding (K neighbors, RBF radius), the MANI process (patch size, initial voxel size, number of iterations, compression rates, similarity/dynamic score weights), spline parameterization (number and selection of keyframes K). Tuning such a large set of parameters can be challenging, raising concerns about the method's applicability and robustness across different types of scenes or datasets. The paper lacks an analysis of how sensitive the final performance is to the choice of these hyperparameters
2.The method heavily relies on pre-trained VFMs for crucial inputs like semantic tokens, depth maps, segmentation masks, and 2D tracklets. Errors or inaccuracies from these upstream models (e.g., poor depth estimation in textureless regions, inaccurate tracking during occlusions, noisy segmentation) could propagate and significantly degrade the quality of node initialization (MANI) and motion supervision.

**Questions:**

1.A key potential benefit of sparse control methods is enabling motion editing. Since MANI initializes nodes based on semantic tokens , do these nodes correspond to meaningful semantic parts? Have the authors explored manipulating these nodes for motion editing or control? Are there any demonstrations showing this capability?
2.Compared to prior work like SC-GS , which primarily relied on geometric node placement, does the semantic nature of your nodes unlock more advanced or more intuitive editing possibilities? For example, could one edit the motion of specific semantic parts (like "arm" or "leg") by manipulating a group of associated nodes?
3.Given the large number of hyperparameters identified in Weakness #1, how were they tuned for the experiments? Is there a recommended strategy for setting these parameters for new scenes? How robust is the performance to variations in key parameters like the MANI compression rates or the number of spline keyframes?

---

> ### Author Response · Authors · 2025-11-28
>
> ### [W1&Q3] Hyperparameter Configuration and Robustness Evaluation
>
> We appreciate the reviewer’s comment. Regarding the hyperparameters listed by the reviewer, we clarify their roles and how they are set in practice.
>
> **First**, several quantities are not free scene-specific knobs. The `RBF radius` is learnable and does not require manual tuning. The MANI `patch size` is tied to the input resolution and is fixed to a small value such as 16 or 32 pixels so that each dynamic region is covered by multiple patches, independent of the underlying geometry. The `initial voxel size` $v_{init}$ is obtained by linearly scaling the scene bounding box so that the first voxel grid yields a reasonable number of candidate anchors. Our iterative compression scheme then reduces the dependence on any particular voxel size. The `target number of anchors` $N_n$ is chosen only at the level of scene type, for example a few hundred nodes for object-centric scenes and a few thousand for real-world scenes. Similar to SC-GS, we perform adaptive anchor insertion and pruning during training, so the final anchor density automatically adjusts to motion complexity and is not overly sensitive to the initial $N_n$. The `number of MANI iterations` is determined by $N_n$ and we simply stop once the anchor count falls below this target. For spline trajectories, the `number of keyframes` $K$ is derived from a fixed `frame interval` $N$ given the sequence length, and we will add a sensitivity study on this keyframe interval.
>
> Second, MANI compression rates and their weights are configured to be adaptive by design. The `per-voxel compression ratio` $r%(C)$ is computed from the dynamic tendency of each voxel, so it does not require manual tuning. The `bounds` $(r _{min}, r _{max})$ control the overall strength of compression. Larger bounds mean that fewer iterations are needed and that dynamic regions keep fewer anchors. Since $r%(C)$ is already adaptive, these bounds have limited influence on the final layout and do not depend strongly on the scene. In our experiments we therefore use a single pair $(r _{min}, r _{max})$ for all scenes, selected by a coarse search on a validation scene, and we will report a sensitivity analysis of these bounds. The `weights` $(α, β, η)$ only balance semantic similarity and foreground masks in the MANI scores and are kept fixed across all scenes; we will also provide an ablation showing that moderate changes of these weights have little effect.
>
> **In summary**, among all the quantities mentioned:
>
> * patch size, initial voxel size $v_{init}$, target anchor number $N_n$, and the number of MANI iterations depend only coarsely on scene scale and type, and our iterative compression together with adaptive density adjustment makes the method robust within a wide range of values;
> *  the RBF radius is learned, and the per-voxel compression ratio $r%(C)$ is computed adaptively, so they do not need to be tuned;
> * the compression bounds $(r _{min}, r _{max})$, MANI weights $(α, β, η)$, and the spline keyframe interval $N$ are only weakly related to individual scenes, and a single global setting works well for all experiments. **We have included additional ablation results for these quantities on Chicken and 3D-Printer Scene on Hyper-NeRF dataset in the revised version and in the table below.**
>
> **Overall**, although the framework involves many parameters, **most of them do not require per-scene retuning**. For the few parameters that are nominally scene-dependent, our deliberately designed mechanisms such as  iterative compression, adaptive anchor density, and dynamic compression rates **reduce the sensitivity of the model to parameter variation**, resulting in broad valid ranges that do not require careful scene-specific tuning.
>
> **Table. Ablation study on $(r _{min}, r _{max})$ combinations.**
>
> |Method|PSNR↑|SSIM↑|LPIPS↓|
> |-|-|-|-|
> |[25,50]|25.87|0.774|0.199|
> |**[25,75]**|**26.10**|**0.796**|**0.196**|
> |[50,75]|26.03|0.790|0.196|
>
> **Table. Ablation study on $η$.**
>
> |Method|PSNR↑|SSIM↑|LPIPS↓|
> |-|-|-|-|
> |0|25.15|0.713|0.236|
> |0.25|25.81|0.768|0.215|
> |**0.5**| **26.10** | **0.796** | **0.196** |
> |0.75|25.95|0.785|0.209|
>
> **Table. Ablation study on $(α, β)$.**
>
> |Method|PSNR↑|SSIM↑|LPIPS↓|
> |-|-|-|-|
> |[0.25,0.75]|25.86|0.778|0.199|
> |[0.5,0.5]|26.05|0.790|0.197|
> |**[0.75, 0.25]**|**26.10**|**0.796**|**0.196**|
> |[1,0]|26.03|0.788|0.197|
>
> **Table. Ablation study on keyframe interval $N$.**
>
> |Method|PSNR↑|SSIM↑|LPIPS↓|FPS↑|
> |-|-|-|-|-|
> |2|25.94|0.782|0.215|101|
> |6|**26.14**|**0.814**|**0.193**| 97     |
> |**8**| 26.10     | 0.796     | 0.196     |90|
> |12|25.88|0.776|0.224|**86**|

---

> ### Author Response · Authors · 2025-11-28
>
> ### [W2] Reliability of VFM Priors
>
> We appreciate the reviewer’s comments on the potential impact of errors from upstream VFMs on our method. In response, we **first** briefly summarize how these priors are used as soft cues rather than hard constraints, **then** point to existing ablations showing that the method does not heavily depend on them, and **finally** add loss term ablations to further support that VFM errors do not significantly degrade node initialization and motion supervision.
>
> * **MANI.** Semantic tokens and foreground masks are used in MANI only to rank node candidates and set voxel level compression ratios, with scores aggregated over local clusters so that noisy or misclassified segments merely perturb this ranking rather than enforcing a fixed node layout. As shown in *Table 3(b)*, replacing MANI with FPS (24.49dB) or voxel based initialization (24.06dB), both of which ignore VFM signals, still yields performance clearly above the baseline (22.35dB), suggesting that when VFM based segmentation is unreliable, node initialization simply falls back to a purely geometric variant instead of failing.
>
> * **MS.** VFM based 2D tracklets are used to initialize spline parameterized node trajectories, which are then fully refined under RGB and ARAP geometric losses. In Table 3(a), "+MS" (24.51dB) and "+MS (w/o Init)" (24.13dB) both clearly outperform the baseline (22.35dB), showing that even without VFM initialization the motion field remains strong, and that inaccurate tracks, for example near occlusions, can be corrected during optimization rather than propagating unbounded errors.
>
> * **Loss.** VFM depth maps and tracking priors enter the loss through moderately weighted depth and tracking terms that serve as auxiliary regularizers on top of strong photometric and geometric supervision. As shown in *Tables 4 and 5*, swapping different VFM depth and tracklet models leads to only minor PSNR changes, indicating that reconstruction quality is insensitive to moderate noise or shifts in these priors, including poor depth in textureless regions and imperfect tracking.
>
> In addition, we will include ablations that disable the depth and tracking losses, showing that even when the corresponding VFM priors are severely degraded, the performance drop is moderate and the method remains stable, effectively degenerating to a sparse control dynamic 3DGS driven mainly by RGB and geometric constraints rather than being dominated by erroneous VFM signals.
>
> **Table. Ablation study on VFM loss terms on Hyper-NeRF dataset.**
>
> | Method          | PSNR↑     | SSIM↑     | LPIPS↓    |
> | --------------- | --------- | --------- | --------- |
> | w/o $L_{mask}$  | 25.46     | 0.691     | 0.259     |
> | w/o $L_{depth}$ | 24.97     | 0.674     | 0.277     |
> | w/o $L_{track}$ | 25.52     | 0.690     | 0.253     |
> | **Ours**        | **25.78** | **0.722** | **0.242** |
>
> ------
>
> ### [Q1&Q2] Semantic Sparse-Control Nodes and Motion Editing
>
> We appreciate the reviewer’s insightful questions about the connection between sparse control and motion editing. Our framework follows the node-based sparse-control design of SC-GS, so in principle the same style of editing operations, such as modifying or freezing selected control nodes, can also be applied to our model without changing the underlying formulation.
>
> On the semantic side, MANI uses semantic tokens and foreground priors as soft cues to bias node initialization toward foreground and motion-salient regions. After optimization, nodes tend to concentrate on these regions, but they are not enforced to form a strict one-to-one mapping to explicit parts such as “arm” or “leg”. This means that the learned nodes remain compatible with more intuitive part-level editing interfaces that group nodes predominantly influencing a given semantic region. We agree that semantically informed motion editing is a very appealing application that can build on top of our sparse-control representation, and we plan to explore this direction in future work.

---

### Official Review · Reviewer_bxh3 · 2025-10-29

**Soundness:** 3
**Presentation:** 4
**Contribution:** 4
**Rating:** 6
**Confidence:** 4

**Summary:**

1. This paper focuses on dynamic 3D reconstruction from monocular videos and proposes a sparse, node-based deformation framework.
2. This paper proposes Motion-Adaptive Node Initialization and spline-parameterized node trajectories with cubic Hermite.
3. Experiments on Hyper-NeRF and N3DV show improved quality and efficiency over previous methods.
4. The idea seems to be novel and the objective experimental results are convincing.

**Strengths:**

1. Motion-aware sparsity allocates more control nodes in motion-heavy regions via semantic/motion priors from vision foundation models, which makes sense and demonstrates a new direction for optimizing 3D/4D reconstruction.
2. Spline-parameterized node trajectories replaces MLP deformation fields for smoother, more stable motion with cubic Hermite initialized from 2D tracklets.
3. They provide enough ablations that isolate contributions of their framework.
4. The objective experimental results are convincing enough to demonstrate the superiority of their method.

**Weaknesses:**

1. In my view, videos are necessary for papers in 4D reconstruction to show the effectiveness and superiority. If there is no videos, there is no enough subjective comparisons. Objective metrics and image/frame comparisons are not enough.
2. This idea of 2D motion-guided deformation is not novel, and many papers have proved the effectiveness. There should be more visualizations and comparisons with such kinds of methods.
3. The method depends heavily on VFM priors. What will happen when there is severe distortions or shifts in the VFM priors? As far as we know, VFM's priors are not that robust and stable so far.

**Questions:**

1. I am curious about the actual effectiveness of trajectory initialization. It fits splines only for translations while starting rotations from the identity. Does it show slow convergence or demand stronger regularization in cases with pronounced rotational or articulated motion?
2. Despite incorporating multiple constraints (RGB, depth, masks,...), are there some scenarios may require stronger geometric priors or multi-view supplementation due to scale and occlusion ambiguities under monocular setups?

---

> ### Author Response · Authors · 2025-11-28
>
> ### [W1] Video Demonstrations
>
> We thank the reviewer’s comments. We have included video results in the supplementary material to better demonstrate the effectiveness and advantages of our method.
>
> ------
>
> ### [W2] Additional Comparisons with 2D Motion-Guided Deformation Methods
>
> We appreciate the reviewer’s comments and have clarified both the comparison and the role of 2D motion in our method. **First**, we **have added experiments in revised version comparing against representative 2D motion-guided approaches (Mosca[1], HiMoR[2], and MotionGS[3])**, and our method consistently achieves higher PSNR and perceptual quality across the evaluated scenes.
>
> **Second**, in our framework 2D tracklets are used to initialize the Hermite spline trajectories of the nodes  so that optimization starts from a reasonable motion guess, and they also provide a soft temporal constraint through the tracking loss $L_{track}$. After this initialization, the trajectories are fully optimized under full objective (Eq. 13). This is structurally different from methods that directly drive deformation by 2D motion cues. Our main novelty therefore lies in motion adaptive node allocation and spline parameterized node trajectories within a sparse 3D Gaussian control framework, rather than in the use of 2D motion itself.
>
> **Finally**, our ablation in *Table 3(c)* and *Figure 5* replaces the proposed MS module with a representative 2D motion-guided variant that relies purely on tracklets ("Tracklet"), which leads to a PSNR drop (25.78 dB -> 24.59 dB) and yields less complete motion reconstructions, which confirms that our design brings clear benefits beyond directly following 2D motion. We will make these points more explicit in the revised manuscript and better clarify the relation and differences between our method and existing 2D motion-guided approaches.
>
> **Table. Additonal baselines on Hyper-NeRF dataset (PSNR↑).**
>
> |Method|Broom|3D-Printer|Chicken|Banana|Mean|
> |-|-|-|-|-|-|
> |MotionGS|22.30|21.80|26.80|28.20|24.78|
> |MoSca|22.14|22.26|28.19|28.43|25.25|
> |**Ours**|22.37|22.53|29.66|28.55|25.78|
>
> **Table. Additonal per-scene comparisons (PSNR↑) on Nvidia[4] dataset.**
>
> |Method|Balloon1|Balloon2|Jumping|Umbrella|Mean|
> |-|-|-|-|-|-|
> |HiMoR|23.90|23.48|20.04|24.30|22.93|
> |MoSca|23.58|27.80|25.01|25.17|25.39|
> |**Ours**|24.39|27.65|25.43|25.69|25.79|
>
> ------
>
> ### [W3] Reliability of VFM Priors
>
> We appreciate the reviewer’s comments on the robustness of VFM priors in our method. In response, we **first** briefly summarize how these priors are used as soft cues rather than hard constraints, **then** point to existing ablations showing that the method does not heavily depend on them, and **finally add loss-term ablations to further support the stability of our framework.**
>
> * **MANI.** semantic tokens and foreground masks are used in MANI to rank node candidates and adjust voxel-level compression ratios, where scores are aggregated over local clusters so that occasional misclassifications mainly change the ordering slightly instead of forcing a specific node layout. *Table 3(b)* in main paper shows that replacing MANI (25.78dB) with FPS (24.49dB) or voxel-based (24.06dB) initialization, which do not use semantic tokens or foreground masks, still yields competitive performance above the baseline (22.35dB).
>
> * **MS.** VFM-based 2D tracklets are used to initialize spline-parameterized node trajectories, but the trajectories are fully optimized later under RGB and ARAP geometric losses; as shown by the "+MS"(24.51dB) vs. "+MS (w/o Init)" (24.13dB) in *Table 3(a)* in main paper, removing this VFM initialization still yields clear gains over the baseline (22.35dB), indicating that the spline motion field does not rely critically on accurate tracklets.
>
> * **Loss.** VFM depth maps and tracking priors also enter the loss through moderately weighted depth and tracking terms, which act as auxiliary regularizers on top of strong photometric and geometric supervision. *Table4 & 5* in Appendix shows that swapping different VFM depth and tracklet models causes only minor PSNR variations, which further indicates that the reconstruction quality is insensitive to moderate shifts in the depth prior.
>
> In addition, we have included new ablations that disable mask, depth and tracklet losses on Hyper-NeRF dataset, showing that even under severely degraded VFM priors the performance decreases only moderately and the method remains stable, effectively degenerating to a sparse-control dynamic 3DGS driven mainly by RGB and geometric constraints.
>
> **Table. Ablation study on VFM loss terms on Hyper-NeRF dataset.**
>
> |Method|PSNR↑|SSIM↑|LPIPS↓|
> |-|-|-|-|
> |w/o $L_{mask}$|25.46|0.691|0.259|
> |w/o $L_{depth}$|24.97|0.674|0.277|
> |w/o $L_{track}$|25.52|0.690|0.253|
> |**Ours**|**25.78**|**0.722**|**0.242**|

---

> > ### Comment · Reviewer_bxh3 · 2025-11-28
> >
> > I am satisfied with the response. I will keep my positive score.

---

### Official Review · Reviewer_mnpE · 2025-11-01

**Soundness:** 3
**Presentation:** 2
**Contribution:** 2
**Rating:** 6
**Confidence:** 3

**Summary:**

This paper proposes a motion-adaptive dynamic 3D reconstruction framework that allocates control points based on motion complexity rather than static geometry. By leveraging semantic and motion priors from vision foundation models and introducing spline-based trajectory parameterization, the proposed approach achieves smoother motion, higher reconstruction quality, and improved efficiency over prior approaches. Extensive experiments demonstrate the effectiveness.

**Strengths:**

1. Leveraging the  vision foundation models for 4D Gaussian Reconstruction seems interesting, though it is not novel.
2. The experiments demonstrate the effectiveness based on two important datasets.
3. Motion-Adaptive Node Initialization seems interesting.

**Weaknesses:**

1. The comparison is slightly old. I feel confused that the paper did not compare directly with [A] and [B].
2. The whole pipeline is quite complex, and this paper did not provide any significant progress.
3. These ideas all make sense, but they’re quite ordinary.

[A] Mosca: Dynamic gaussian fusion from casual videos via 4d motion scaffolds
[B] Himor: Monocular deformable gaussian reconstruction with hierarchical motion representation.

**Questions:**

n/a

---

### Meta-Review · Area_Chair_UjwP · 2026-01-06

**Summary:**

During the rebuttal phase, the authors successfully addressed the majority of the reviewers' concerns through extensive additional experiments and clarifications. Specifically, they added missing baseline comparisons (including MoSca, HiMoR, and MotionGS) to demonstrate the method's competitiveness. They also provided video demonstrations to supplement the qualitative results. Furthermore, through detailed ablation studies regarding VFM loss terms and hyperparameters, the authors effectively countered concerns about the method's potential over-reliance on pre-trained model priors (mask, depth, tracking) and sensitivity to hyperparameter tuning, showing the system remains stable even under degraded conditions .

However, some issues still exist (e.g., visual quality v.s. system complexity, novelty, ...)

Personally, I find the methodology interesting and the paper logic easy to follow. The work makes a solid contribution to the field of 4D reconstruction based on sparse control points. The authors' approach of using semantic tokens from Vision Foundation Models to guide node initialization (MANI) is a clever solution to the mismatch between geometric allocation and motion complexity. The authors demonstrate strong implementation skills. Thus I would recommend acceptance.

Minor issues:
It seems almost all the citations are in the wrong format (not in parentheses).

**Reviewer Concerns:**

See above.

**Reviewer Scores:**

Considering the extensive and convincing experiments in the rebuttal, I would guess reviewers would like to at least maintain the scores and perhaps increase the scores a little bit.

---

### Decision · Program_Chairs · 2026-01-26

Accept (Poster)